# Hyperglycemia in Pet African Pygmy Hedgehogs (*Atelerix albiventris*): Prevalence, Clinical Characteristics, and Prognostic Indicators of Severe Hyperglycemia

**DOI:** 10.3390/ani15101455

**Published:** 2025-05-18

**Authors:** Do-Hyun Kwak, Myung-Chul Kim, Woo-Jin Song, Young-Min Yun

**Affiliations:** 1College of Veterinary Medicine, Veterinary Medical Research Institute, Jeju National University, Jeju 63243, Republic of Korea; happycare719@naver.com (D.-H.K.); mck@jejunu.ac.kr (M.-C.K.); ssong@jejunu.ac.kr (W.-J.S.); 2Sangdo Healing Animal Hospital, 353 Sangdo-ro, Dongjak-gu, Seoul 06977, Republic of Korea

**Keywords:** African pygmy hedgehog, blood glucose, hyperglycemia, mortality, prevalence, prognosis, prognostic indicators, stress hyperglycemia

## Abstract

Hyperglycemia is known to negatively affect patients in both humans and various animal species. However, information on hyperglycemia in hedgehogs is extremely limited. This study investigated 252 African pygmy hedgehogs and 579 blood test results collected over a three-year period from a veterinary hospital. The frequency of hyperglycemic samples among all blood tests was 48.1% (278/579). Twenty-eight hedgehogs were identified with severe hyperglycemia, defined as blood glucose levels of 180 mg/dL or higher. The prevalence of severe hyperglycemia in hedgehogs was 11.1% (28/252). The most common clinical signs in these hedgehogs were anorexia and decreased activity. The mortality rate among these hedgehogs was 53.6% (15/28). Mortality in hedgehogs with severe hyperglycemia was significantly associated with the severity of anorexia on the day severe hyperglycemia was first identified. Additionally, recovery from severe hyperglycemia on the next blood test was associated with a favorable prognosis. These potential prognostic indicators may assist in the early prediction of outcomes and in determining clinical management strategies for hedgehogs with severe hyperglycemia.

## 1. Introduction

Hyperglycemia has been reported in many animals, including dogs, cats, horses, ferrets, raccoons, rabbits, guinea pigs, chinchillas, birds, bearded dragons, and iguanas [1,2,3,4,5,6,7,8,9,10]. The causes and mechanisms of hyperglycemia are complex and involve multiple factors, including stress, diseases such as pancreatitis and glucagonoma, fear, increased insulin resistance, hormonal abnormalities, medications like exogenous steroids, and diabetes mellitus [11].

In small mammals, blood glucose levels are known to be influenced by a variety of physiological factors. In particular, fasting duration, the composition of the most recent meal, the level of stress during anesthesia [12,13,14], body condition score, sex and neuter status [15], and, in females, estrus stage or pregnancy status [16] have all been reported as important factors affecting glucose homeostasis.

The term stress hyperglycemia is used to describe a condition where hyperglycemia is induced by illness or fear in patients without diabetes. This phenomenon is observed in both humans and animals, including dogs, cats, and horses [17,18,19,20]. In veterinary medicine, stress hyperglycemia is considered not only a physiological response but also a potential prognostic indicator of disease outcomes.

Many studies have suggested that hyperglycemia is associated with disease severity and poor prognosis in animals with various diseases [1,18,21,22,23,24,25,26,27]. In dogs and cats with head trauma, the severity of hyperglycemia was significantly associated with the degree of neurologic injury [1]. Hyperglycemia was associated with a worse prognosis in horses with acute gastrointestinal disease [18]. In a study of hospitalized dogs, the blood glucose levels of dogs that died were found to be higher than those of dogs that survived [21]. In dogs with congestive heart failure, low blood sodium and high plasma glucose were associated with poor patient outcomes [22]. In rabbits, severe hyperglycemia was associated with poor prognosis, and hyperglycemia served as a measurable parameter to differentiate between gut stasis and intestinal obstruction in rabbits with anorexia [23]. The negative effects of hyperglycemia have been studied in many animals, but reports on the clinical impact of hyperglycemia in hedgehogs are lacking.

The reference interval for serum glucose levels in hedgehogs has been established [28,29,30,31]. However, hyperglycemia was not investigated in these studies. There is a report that mentions hyperglycemia in African pygmy hedgehogs (*Atelerix albiventris*) [32]. This report described hyperglycemia and the detection of glucose in urine during the treatment of a hedgehog with mammary adenocarcinoma and a granulosa cell tumor. However, since it was a case report on a single hedgehog, the information on the clinical impact and prevalence of hyperglycemia remains limited.

The first objective of this study is to investigate the overall frequency of hyperglycemia in blood samples collected from hedgehogs. The second objective is to determine the blood glucose range in healthy hedgehogs and compare it with known reference values. The third objective is to examine the pathophysiological information, prevalence, mortality rates, and prognostic factors associated with severe hyperglycemia in hedgehogs.

## 2. Materials and Methods

The medical records of African pygmy hedgehogs that visited a local veterinary hospital from January 2019 to December 2021 were retrospectively examined. During this period, there were 2388 medical records and 664 blood test records from 333 hedgehogs. Among these, 579 blood samples that included blood glucose values and the medical records of 252 hedgehogs who underwent blood glucose testing were included in this study. All the hedgehogs were privately owned pets that lived indoors.

### 2.1. Classification Criteria for All Blood Samples

To evaluate the frequency and distribution of hyperglycemia, all blood samples with recorded bloodglucose values were included in the analysis. Based on a previously established reference range (89 ± 30 mg/dL) [28], all blood samples were categorized into four groups according to their glucose levels: a hypoglycemia group (<59 mg/dL), a normoglycemia group (59–119 mg/dL), a mild hyperglycemia group (120–179 mg/dL), and a severe hyperglycemia group (≥180 mg/dL).

### 2.2. Healthy Hedgehog Criteria

A separate group of healthy hedgehogs was defined for comparison with a previously reported reference range (89 ± 30 mg/dL) [28]. Hedgehogs were included in this group if they visited the hospital for routine health check-ups, showed good activity and appetite, and had no abnormalities on physical examination. Individuals that had recently received medication, exhibited clinical signs of disease, or had physical deformities were excluded. Blood glucose level, age, sex, and body weight were recorded at the time of examination. In cases where multiple health examinations were performed, data from the first examination were used. This group was defined independently and was not used for direct comparison with hedgehogs with severe hyperglycemia.

### 2.3. Severe Hyperglycemic Hedgehog Criteria

Body weight, age, sex, clinical symptoms, and underlying diseases in hedgehogs with blood glucose concentrations ≥180 mg/dL were investigated. In cases with multiple test results, only the data from the first day on which severe hyperglycemia was confirmed were used. Hedgehogs that survived more than 100 days after severe hyperglycemia was confirmed were classified as survivors. Those that died within 100 days were classified as non-survivors, and their date of death was recorded. Additionally, medical records were reviewed to explore possible prognostic factors associated with mortality in hedgehogs with severe hyperglycemia. The evaluated variables included appetite status on the first day severe hyperglycemia was confirmed and recovery from severe hyperglycemia. Hedgehogs were categorized into three groups based on the degree of appetite loss: the Good appetite group, which showed normal food intake; the Decreased appetite group, which showed a noticeable reduction in food intake but continued to eat; and the Complete anorexia group, in which no food intake was observed for more than 24 h. In addition, hedgehogs were classified into two groups based on recovery from severe hyperglycemia, and 100-day mortality rates were compared between the recovery and non-recovery groups.

### 2.4. Laboratory Examination

All blood collection and physical examinations were performed under general inhalation anesthesia using isoflurane (Ifran Liq., Hana Pharm, Inc., Seoul, Republic of Korea). All hedgehogs were placed in a small induction chamber and sedated within 2 min using 5% isoflurane in 5 L/min oxygen. Subsequently, anesthesia was maintained via a mask using 3% isoflurane in 1 L/min oxygen. Blood collection was performed immediately after sedation and completed within 5 min of anesthesia induction. A volume of 0.5 to 0.8 mL of blood was collected from the jugular vein or the cranial vena cava. Maintenance concentrations were adjusted according to individual clinical parameters such as respiratory rate and depth. Blood samples for biochemical analyses were collected immediately into heparinized tubes and tested within 10 min. Blood glucose measurements and serum biochemical tests were performed using DRI-CHEM NX 500 (Fujifilm Corporation, Tokyo, Japan). CBC, electrolytes, X-rays, ultrasound, dipstick, cytology, and biopsy were selectively performed based on the purpose of the visit and the condition of the hedgehog. All procedures, including blood sampling, were conducted with owner consent as part of routine clinical care.

### 2.5. Statistics

The means, standard deviations (SDs), medians, and confidence intervals (CIs) were calculated for each group. Independent *t*-tests were used to compare continuous variables between two groups (e.g., survivors vs. non-survivors), and were conducted using SciPy v1.14.1 in Python v3.11.5. Categorical variables were analyzed using Fisher–Freeman exact tests. These were primarily performed using SciPy, but for contingency tables with more than three categories, *p*-values were additionally computed using a Monte Carlo simulation-based test in R v4.3.0 due to limitations in Python’s support for larger tables. A value of *p* < 0.05 was considered statistically significant.

## 3. Results

To improve clarity, we note that the results are structured around three separate datasets, each with unique inclusion criteria and analytical purposes. As such, they represent independent study groups and are not intended for direct intergroup comparison.

### 3.1. Blood Glucose Level Classification

Among the 579 blood glucose test results, 6 were in the hypoglycemia group (<59 mg/dL), 295 in the normoglycemia group (59–119 mg/dL), 211 in the mild hyperglycemia group (120–179 mg/dL), and 67 in the severe hyperglycemia group (≥180 mg/dL), as shown in Table 1. The overall frequency of hyperglycemia (≥120 mg/dL) among all samples was 48.1% (278/579). The mean blood glucose level across all samples was 137.8 ± 73.7 mg/dL (range, 28–586).

### 3.2. Healthy Hedgehog Group

Of the 252 hedgehogs, 80 met the inclusion criteria for the healthy group. The sex distribution, age, body weight, and blood glucose levels of these hedgehogs are summarized in Table 2. The mean age was 17.4 months, and the average body weight was 496.7 g. The mean blood glucose level of healthy pet African pygmy hedgehogs was 110.4 ± 16.7 mg/dL (range, 72–153), as shown in Table 2. These values were obtained to compare the blood glucose levels of healthy pet hedgehogs in this study with the previously reported reference range (89 ± 30 mg/dL) [28]. This group was not designed as a comparison group for hedgehogs with severe hyperglycemia.

### 3.3. Severe Hyperglycemia Hedgehog Group

The prevalence of severe hyperglycemia among the 252 hedgehogs was 11.1% (28/252). The 28 hedgehogs with severe hyperglycemia included 15 intact males, 2 neutered males, 7 intact females, and 4 spayed females. The mean age at the time of severe hyperglycemia was 36.8 months (range, 8.8–65.1 months), and the average body weight was 569.3 ± 161.2 g (range, 237–876 g). The mean body weight of the 17 male hedgehogs was 546.6 ± 157.4 g; the mean body weight of the 11 female hedgehogs was 604.4 ± 168.3 g. On the first day when severe hyperglycemia was confirmed, the mean blood glucose level was 269.1 ± 81.6 mg/dL (range, 186–522), as shown in Table 3.

Of the hedgehogs with severe hyperglycemia, 24 had anorexia and 23 had a loss of activity. Seventeen patients had stool abnormalities (green stool, soft stool, melena, and watery diarrhea), and four had urinary abnormalities (dysuria and green urine). Thirteen hedgehogs had recently undergone surgery for tumor removal or had tumors. Eleven hedgehogs exhibited neurologic signs, including gait abnormalities such as tetraplegia, bilateral hind limb paresis, bilateral forelimb paresis, and circling associated with torticollis. Of these, bilateral hind limb paresis was the most common (*n* = 6). Four patients were bilaterally blind. Other symptoms such as collapse, vomiting, jaundice, cardiomegaly, pulmonary edema, and temporary increases in drinking water were confirmed (Table 4). Many of the hedgehogs exhibited multiple concurrent clinical signs. Among the 28 hedgehogs with severe hyperglycemia, only 1 was asymptomatic and 1 exhibited a single clinical sign. In total, 26 out of 28 hedgehogs (92.9%) had two or more clinical signs, 25 (89.3%) had three or more, 20 (71.4%) had four or more, 13 (46.4%) had five or more, and 4 (14.3%) had six or more. No hedgehogs exhibited more than six clinical signs. Among the 26 hedgehogs with two or more signs, the most common combination was anorexia + loss of activity, observed in 21 hedgehogs. The second most frequent combinations were anorexia + abnormal feces and loss of activity + abnormal feces, each identified in 16 hedgehogs. In the 25 hedgehogs with three or more clinical signs, the most common triad was anorexia + loss of activity + abnormal feces, noted in 15 hedgehogs.

Fifteen of the twenty-eight hedgehogs died within 100 days after severe hyperglycemia was confirmed, with a mortality rate of 53.6% (15/28). The average survival time of the deceased hedgehogs was 15.9 days (range, 0–46 days). The remaining 13 hedgehogs survived for more than 100 days after severe hyperglycemia was confirmed (Table 5).

Twelve of twenty-eight animals recovered from hyperglycemia. The average recovery period was 14.4 days (range, 7–30 days). The remaining 16 animals did not recover from hyperglycemia (Table 5). In two hedgehogs, severe hyperglycemia improved as a result of subcutaneous injection of insulin (Humulin N Injection 100 unit, Lily Korea, Inc., Seoul, Republic of Korea) at a dose of 0.2 to 2 U/kg every 12 h. These two hedgehogs had their blood glucose levels stabilized by insulin administration but returned to a hyperglycemic state within 12 h. With repeated long-term insulin treatment, the two hedgehogs survived for more than 100 days (250 and 343 days, respectively).

The 28 hedgehogs with severe hyperglycemia were categorized according to the severity of appetite loss on the first day severe hyperglycemia was confirmed, as defined in the methods. They were classified as survivors or non-survivors based on whether they died within 100 days (Table 6). The mortality rate was 0% (0/4) in the Good appetite group, 46% (6/13) in the Decreased appetite group, and 82% (9/11) in the Complete anorexia group. The greater the severity of anorexia on the first day severe hyperglycemia was confirmed, the higher the mortality rate (*p* = 0.013).

Among the 28 hedgehogs, follow-up blood glucose tests were conducted an average of 14.6 days after the initial diagnosis (range, 3–30 days), during which 12 animals were confirmed to have recovered from severe hyperglycemia. The average time to confirmed recovery among these 12 hedgehogs was 14.4 days (range, 7–30 days). The mortality rate was 8.3% (1/12) in the recovery group and 87.5% (14/16) in the non-recovery group (Table 7). Hedgehogs that had recovered from severe hyperglycemia by the time of follow-up testing had a significantly lower risk of death within 100 days (*p* < 0.001).

The mean age and body weight significantly differed between survivors (*n* = 13) and non-survivors (*n* = 15) among hedgehogs with severe hyperglycemia. Non-survivors were on average older (*p* = 0.008) and had lower body weight (*p* = 0.032) compared to survivors. In contrast, no significant difference was observed in the initial blood glucose concentration between the two groups (*p* = 0.523), as shown in Table 8.

## 4. Discussion

In this retrospective study, 579 blood glucose test results were investigated in 252 hedgehogs who visited a local veterinary hospital over a period of three years. Additionally, blood glucose levels were investigated in 80 healthy hedgehogs and 28 hedgehogs with severe hyperglycemia. To the best of our knowledge, this is the first study to focus on hyperglycemia in African pygmy hedgehogs (*Atelerix albiventris*).

Many studies have reported the prevalence of hyperglycemia in various animal species. The prevalence of hyperglycemia in sick cats was reported to be 34% [33], and another study reported the prevalence of hyperglycemia in hospitalized cats to be 65% [26]. The prevalence of hyperglycemia in dogs admitted to intensive care units was reported to be 16% [21], and the prevalence of hyperglycemia in sick rabbits was reported to be 37% [34]. In our study, the prevalence of severe hyperglycemia among the 252 hedgehogs examined was 11.1% (28/252). The overall frequency of hyperglycemia among all blood samples was 48.1% (278/579), with mild and severe hyperglycemia observed in 36.5% (211/579) and 11.6% (67/579) of samples, respectively. The frequency of hyperglycemia in hedgehogs observed in this study (48.1%) was higher than expected. This may be attributed to the overrepresentation of samples from sick animals in the blood sample pool. To further investigate this possibility, a subgroup analysis was performed on hedgehogs that had undergone five or more blood glucose measurements. Among the 252 hedgehogs included in this study, 20 met this criterion, and a total of 193 blood glucose results were obtained from these individuals. Among these, the distribution of blood glucose levels was as follows: hypoglycemia in 1.0% (2/193), normoglycemia in 44.0% (85/193), mild hyperglycemia in 32.1% (62/193), and severe hyperglycemia in 22.8% (44/193). The frequency of hyperglycemia in this subgroup was 54.9% (106/193), which is higher than the overall frequency of 48.1% (278/579) observed in all samples. These findings suggest that the high frequency of hyperglycemia observed in this study may be partly explained by more frequent hospital visits and repeated blood sampling in sick hedgehogs. In this context, a previous study discussed possible reasons for the high frequency of hyperglycemia. That study reported that the annual increase in the frequency of stress-induced hyperglycemia in cats was not due to a true increase in hyperglycemia itself, but rather to a rise in the number of cats visiting animal hospitals and an increase in the frequency of blood testing [20]. In this study, the high frequency of hyperglycemia among all blood samples (48.1%) suggests that veterinarians may frequently encounter hyperglycemia during the clinical care of hedgehogs. The frequency of severe hyperglycemia was 11.6%, indicating that veterinarians may occasionally encounter severe hyperglycemia during the clinical care of hedgehogs. This rate of occurrence emphasizes the importance of this study on severe hyperglycemia in hedgehogs.

In the present study, a separate group of 80 healthy African pygmy hedgehogs was selected to assess blood glucose levels. The mean blood glucose level was 110.4 ± 16.7 mg/dL (range, 72–153), which was higher than previously reported values for this species. A previous study reported a mean blood glucose level of 89 ± 30 mg/dL in African pygmy hedgehogs [28]. Another study of 23 wild African pygmy hedgehogs reported a mean of 86.05 ± 4.65 mg/dL (range, 60–125) [29]. The glucose values in that study were labeled as fasting glucose. The higher glucose levels observed in our study compared to these may be attributable to differences in fasting status. Other factors may also account for the differences in blood glucose levels. Wild hedgehogs may face challenges in hunting, which could result in lower blood glucose levels. In contrast, pet hedgehogs have access to an abundant food supply. Furthermore, transportation stress to the veterinary hospital may lead to elevated blood glucose levels in pet hedgehogs. In the present study, isoflurane was used as an anesthetic agent. Previous reports have demonstrated that isoflurane can affect blood glucose levels in various animal species [12,13,14,35,36,37]. Although the effect of isoflurane on glucose metabolism in hedgehogs has not been clearly established, its potential influence should be taken into account. A recent study reported a mean blood glucose level of 108.5 ± 24.1 mg/dL in 37 healthy European hedgehogs (*Erinaceus europaeus*) [30]. This result is very similar to the blood glucose range observed in healthy hedgehogs in the present study. However, this retrospective study [30] did not provide information regarding fasting status. Another study reported a mean blood glucose level of 106.2 ± 14.4 mg/dL (range, 77.4–131.4 mg/dL) in 30 rehabilitated wild European hedgehogs housed in wildlife centers [31], which is also similar to the findings of our study. In that study [31], the authors noted that fasting prior to blood collection could not be controlled due to the nature of wildlife rehabilitation. In our retrospective study, fasting prior to blood sampling could not be controlled. This should be taken into consideration when interpreting our glucose values. Nevertheless, the results of this study are meaningful in that they provide a reference range for blood glucose levels in healthy pet hedgehogs, for which baseline data are currently limited.

Various studies have reported that physical abnormalities, pain, and underlying diseases can induce stress-related hyperglycemia in animals. In cattle with mucosal disease, the prevalence of hyperglycemia was notably high at 86.4% (133/154) [38], while in cases of hemorrhagic bowel syndrome, 18 out of 19 affected cattle exhibited hyperglycemia [39]. In dogs with fractures, severe hyperglycemia (up to 439 mg/dL) was attributed to pain-induced stress [40]. Additionally, hyperglycemia is frequently observed in dogs and cats with head trauma, and the severity of hyperglycemia has been shown to significantly correlate with the degree of neurologic injury [1]. Physical abnormalities and certain diseases have been reported to induce hyperglycemia in various animal species, suggesting that similar factors may contribute to the development of hyperglycemia in hedgehogs. In the present study, the most frequently observed clinical signs in hedgehogs with severe hyperglycemia were anorexia (*n* = 24), reduced activity (*n* = 23), and abnormal feces (*n* = 17). The most commonly identified underlying conditions were tumors (*n* = 13) and neurologic signs (*n* = 11) (Table 4). These findings indicate that tumors and neurologic abnormalities may be potentially involved in the development of stress-related hyperglycemia in hedgehogs. However, because the occurrence of hyperglycemia is influenced by multiple factors, it is difficult to establish a causal relationship with any specific disease. Therefore, to better understand and confirm these associations, further studies, including histopathological analysis, are warranted.

Hyperglycemia can be influenced by a variety of factors. In this study, most hedgehogs with severe hyperglycemia were fed standard commercial hedgehog or cat food, and no notable dietary differences were observed. Although no statistically significant difference in sex distribution was found between the healthy and severely hyperglycemic groups (*p* = 0.84), this result should be interpreted with caution, as the healthy group was not randomly selected. Therefore, the findings related to sex distribution are for reference only and should not be considered conclusive. Obesity is a major contributor to insulin resistance, which, when accompanied by β-cell dysfunction, may lead to hyperglycemia [41,42]. In this study, the mean body weight of hedgehogs with severe hyperglycemia was 569.3 ± 161.2 g. The mean body weight of 17 male hedgehogs was 546.6 ± 157.4 g, which is consistent with previously reported normal male weights (500–600 g) for this species [43]. In contrast, the mean body weight of 11 female hedgehogs was 604.4 ± 168.3 g, exceeding the previously reported normal range for females (250–400 g) [43], suggesting a tendency toward obesity. Although female hedgehogs showed a tendency toward obesity, obesity does not appear to negatively affect the prognosis of hedgehogs with severe hyperglycemia. The non-survivors who died within 100 days tended to have lower body weights than the survivors (*p* = 0.032). This finding suggests that lower body weight, rather than obesity, may be associated with poorer survival, and that obesity may not be a negative prognostic factor in hedgehogs with severe hyperglycemia. Indeed, older age and lower body weight are well-established risk factors for poor prognosis in many species [44,45]. However, due to the limited information available on hedgehogs, these results should be interpreted with caution. Further studies are needed to determine the clinical significance of these findings in this species.

Numerous studies have reported that stress-induced hyperglycemia caused by underlying disease can serve as a prognostic indicator for poor outcomes in animals [18,21,22,23,46]. In horses with acute colic, hyperglycemia was associated with decreased survival at hospital discharge, suggesting that elevated blood glucose levels may reflect disease severity and be useful in early prognostic assessment [18]. A study on critically ill dogs found that hyperglycemia was linked to prolonged hospitalization and increased mortality, reinforcing the clinical value of measuring blood glucose concentrations at admission [21]. Similarly, in dogs with congestive heart failure, both hyponatremia and hyperglycemia prior to treatment were associated with poor outcomes, indicating the importance of assessing these parameters during initial evaluation [22]. In pet rabbits, hyperglycemia has been linked to stress and severe illness, and blood glucose monitoring has been proposed as a useful prognostic tool, particularly in cases involving anorexia or emergency presentations [23]. In an emergency room study involving dogs, abnormal blood glucose levels measured upon presentation were also found to be associated with illness severity and clinical outcome, further supporting the prognostic relevance of this parameter [46]. While the impact of hyperglycemia has been investigated in various animal species, its clinical significance in hedgehogs remains unknown. In the present study, hedgehogs with severe hyperglycemia (mean blood glucose level: 269.1 ± 81.6 mg/dL) had a 100-day mortality rate of 53.6%, indicating a poor prognosis. The average time to death in these cases was 15.9 days (range: 0–46 days), highlighting the rapid clinical decline often associated with severe hyperglycemia. These findings suggest that marked hyperglycemia may also serve as a negative prognostic indicator in African pygmy hedgehogs. However, because this study was conducted in a retrospective manner without control over disease type or clinical context, caution is warranted when interpreting the results.

Assessing the severity of anorexia on the first day of diagnosis of severe hyperglycemia may be a potential predictor of prognosis in hedgehogs. The results of this study showed that the mortality rate was higher when the severity of anorexia was greater on the day of examination (*p* = 0.013). These results suggest that the severity of anorexia on the day when severe hyperglycemia was first identified can be used as a useful early warning signal to predict the prognosis of hedgehogs. Appetite is a representative factor reflecting the severity of systemic diseases [47,48]. Restricting food intake is generally associated with maintaining blood glucose or lowering blood glucose. However, in severe diseases or pains that cause a decrease in appetite or complete loss of appetite, the hyperglycemic mechanism may be activated and blood glucose may rise even higher [17,21,49]. In one study, all cases of hyperglycemia exceeding 360 mg/dL in rabbits with complete anorexia were in the presence of serious, life-threatening diseases. In cases of complete anorexia, high blood glucose levels could be used as a predictor to differentiate between intestinal obstruction and intestinal stasis (gut stasis mean 153 mg/dL; intestinal obstruction mean 445 mg/dL) [23]. Another study suggested that anorexia may be a prognostic factor in rabbits with high BUN [50]. Patients with severe hyperglycemia and anorexia may be more likely to have a higher severity of disease and should be managed as patients requiring more aggressive medical intervention. Although not perfect in all patients, this early screening indicator may be a useful reference for clinicians in determining patients with severe disease. Assessing the severity of anorexia is practical for clinicians, as it is an easily observable factor. It may be particularly useful in animals where it is difficult to assess the patient’s condition, such as hedgehogs.

In this study, the mean interval between the initial examination and the follow-up test was 14.6 days (range, 3–30). Because hedgehogs have a small body size and limited blood volume, frequent blood sampling can adversely affect their health [28,51]. Therefore, most follow-up blood tests were performed within 7–30 days. In two exceptional cases, follow-up tests were performed three days after the initial diagnosis at the owner’s request. Among the 12 hedgehogs that recovered from severe hyperglycemia, the average time to confirmed recovery was 14.4 days, similar to the typical follow-up interval. However, the actual time to recovery from severe hyperglycemia may have been shorter than the observed mean of 14.4 days, given the relatively long intervals between tests.

At the follow-up performed after a mean of 14.6 days, hedgehogs that had recovered from severe hyperglycemia showed a significantly lower mortality rate (*p* < 0.001). The mortality rate was 8.3% (1/12) in hedgehogs that recovered, compared with 87.5% (14/16) in those that did not. Considering these results, determining recovery from severe hyperglycemia at follow-up may serve as a potential prognostic indicator. However, this indicator should be used only as a limited and supportive tool. Although hyperglycemia is associated with disease severity in certain conditions [1] and may serve as a predictor of poor outcomes [18,23,46], normalization of blood glucose alone cannot be assumed to improve survival across all types of disease [52]. Therefore, when evaluating prognosis, intuitive clinical signs—such as improved appetite, overall condition, and resolution of presenting symptoms—should also be considered. Due to the relatively long interval between initial and follow-up blood tests in this study, the most effective timing for follow-up testing to evaluate prognosis could not be determined. Further research is needed to clarify this. Despite these limitations, the markedly poor prognosis observed in patients with persistent severe hyperglycemia suggests that this indicator may still have potential for clinical application.

Due to the limitations of a retrospective study, fasting prior to the blood test could not be confirmed in the medical records of this study. This factor may affect blood glucose levels. Therefore, a limitation of this study is that the blood glucose levels cannot be confidently identified as fasting values. In general, wild hedgehogs are nocturnal animals and tend to eat primarily at night [53]. Although all the hedgehogs in this study visited the veterinary hospital during the daytime, it should be cautiously noted that pet hedgehogs may eat intermittently throughout the day and night. However, 24 (86%) of the 28 hedgehogs with severe hyperglycemia exhibited a marked decrease in appetite. Therefore, it is unlikely that the blood glucose levels of the severely hyperglycemic hedgehogs in this study were influenced by the postprandial state.

Several previous studies excluded diabetic patients when investigating hyperglycemia [20,21,26,33]. Two hedgehogs in this study were in a state of persistent hyperglycemia, and it was confirmed that blood glucose levels decreased in response to insulin treatment. These two hedgehogs survived for more than 100 days (250 and 343 days, respectively) while receiving treatment for their hyperglycemic condition. These hedgehogs may have developed diabetes mellitus. However, because diabetes has not yet been reported in hedgehogs, these patients cannot be excluded from the hyperglycemic group. Further studies involving larger datasets and similar case collections are needed to confirm the diagnosis of diabetes mellitus in hedgehogs and to establish appropriate treatment strategies.

## 5. Conclusions

In conclusion, this study identified the prevalence of severe hyperglycemia in pet hedgehogs, which had not been previously reported, and also determined the frequency of hyperglycemia in all collected blood samples. Assessing the severity of anorexia on the first day when severe hyperglycemia is confirmed in hedgehogs can be used as a potential predictor of prognosis. In addition, confirming recovery from severe hyperglycemia at the follow-up examination may serve as a potential and supportive prognostic indicator in hedgehogs. The findings of this study may serve as valuable baseline data for the relatively early assessment of prognosis and the selection of treatment options in the future management of hyperglycemic hedgehogs.

## Figures and Tables

**Table 1 animals-15-01455-t001:** Classification and blood glucose levels of 579 blood samples.

Group	*n*	%	Mean ± SD	Median	Range	95% CI
**Hypoglycemia**	6	1	44.5 ± 13.4	50	28–56	33.8–55.2
**Normoglycemia**	295	50.9	100.5 ± 12.7	103	61–119	99.1–101.9
**Mild hyperglycemia**	211	36.5	138.5 ± 14.7	134	120–178	136.6–140.4
**Severe hyperglycemia**	67	11.6	308.6 ± 99.1	276	181–586	284.9–332.3
**Total**	579	100	137.8 ± 73.7	118	28–586	131.8–143.8

*n*, number of hedgehogs; SD, standard deviation; CI, confidence interval.

**Table 2 animals-15-01455-t002:** Blood glucose levels and physical information of 80 healthy hedgehogs.

	Unit	*n*	Mean ± SD	Median	Range	95% CI
**Age**	month	80	17.4 ± 10.6	15	4.4–50.5	15.1–19.7
**Body weight**	G	80	496.7 ± 146.1	483.5	258–825	464.7–528.7
**Blood glucose**	mg/dL	80	110.4 ± 16.7	111.5	72–153	106.7–114.1
**Sex**	IM NM	39 4				
	IF SF	27 10				

*n*, number of hedgehogs; SD, standard deviation; CI, confidence interval; IM, intact male; NM, neutered male; IF, intact female; SF, spayed female.

**Table 3 animals-15-01455-t003:** Blood glucose levels and physical information of 28 hedgehogs with severe hyperglycemia.

	Unit	*n*	Mean ± SD	Median	Range	95% CI
**Age**	month	28	36.8 ± 17.6	32.7	8.8–65.1	30.3–43.4
**Body weight**	G	28	569.3 ± 161.2	565.5	237–876	509.6–629
**Blood glucose**	mg/dL	28	269.1 ± 81.6	251.5	186–522	238.9–299.4
**Sex**	IM NM	15 2				
	IF SF	7 4				

*n*, number of hedgehogs; SD, standard deviation; CI, confidence interval; IM, intact male; NM, neutered male; IF, intact female; SF, spayed female.

**Table 4 animals-15-01455-t004:** Clinical conditions and underlying diseases of 28 hedgehogs with severe hyperglycemia.

Clinical Conditions	Overall 28 (100%)	
	*n*	%
Anorexia	24	85.7
Loss of activity	23	82.1
Abnormal feces	17	60.7
Neoplasia	13	46.4
Neurologic signs	11	39.3
Bilateral blindness	4	14.3
Abnormal urine	4	14.3
Vomiting	4	14.3
Cardiomegaly	3	10.7
Pulmonary edema	3	10.7
Distal limb disorders	3	10.7
Collapse	3	10.7
Severe IASG hypertrophy *	1	3.6
Jaundice	1	3.6
Increased water intake	1	3.6

* IASG, internal accessory sex gland.

**Table 5 animals-15-01455-t005:** Prognosis of 28 hedgehogs with severe hyperglycemia.

Prognosis	*n*	%
Hyperglycemia recovery *	12	42.9
Hyperglycemia non-recovery	16	57.1
Total	28	100
Survivors (more than 100 days)	13	46.4
Non-survivors (within 100 days)	15	53.6
Total	28	100

* Two of the twelve hedgehogs received insulin treatment and survived more than 100 days (survived 250 days and 343 days).

**Table 6 animals-15-01455-t006:** Prognosis according to the severity of anorexia on the day severe hyperglycemia was confirmed in 28 hedgehogs.

	*n*	Survivors	Non-Survivors	Mortality (%)
Good appetite	4	4	0	0
Decreased appetite	13	7	6	46
Complete anorexia	11	2	9	82
Total	28	13	15	53.6

**Table 7 animals-15-01455-t007:** Prognosis of hedgehogs according to recovery from severe hyperglycemia.

	*n*	Survivors	Non-Survivors	Mortality (%)
Recovery	12	11	1	8.3
Non-recovery	16	2	14	87.5
Total	28	13	15	53.6

**Table 8 animals-15-01455-t008:** Comparison of patient characteristics between survivors and non-survivors (severe hyperglycemia group).

	Survivors (*n* = 13) Mean ± SD	Non-Survivors (*n* = 15) Mean ± SD	*p*-Value (*t*-Test)
First glucose (mg/dL)	258.3 ± 93.5	278.5 ± 71.9	0.523
Body weight (g)	645.2 ± 158.5	503.5 ± 136.4	0.032 *
Age (months)	27.7 ± 14.6	44.6 ± 16.5	0.008 *

* Statistically significant at *p* < 0.05.

## Data Availability

The raw data that were analyzed in this article are available upon direct request to the authors.

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
