# Peer review of "Hyperglycemia in Pet African Pygmy Hedgehogs (Atelerix albiventris): Prevalence, Clinical Characteristics, and Prognostic Indicators of Severe Hyperglycemia"

_animals, 2025, doi:10.3390/ani15101455_

Round 1

Reviewer 1 Report

Comments and Suggestions for Authors

This manuscript describes a retrospective review of glucose levels in healthy and hyperglycemic pet hedgehogs. The information regarding the glucose levels is interesting, and publishing this would help exotic/pocket pet veterinarians particularly with the possibility of prognostic markers. However, the comparison of healthy vs severely hyperglycemic animals confuses me. Healthy vs. sick, with rates of glucose categories makes sense, or an exploration of the differences in normo- vs hyperglycemic (e.g. X number of hedgehogs were healthy with hyperglycemia). Additionally, results are presented in the text of the results section, repeated in the tables, and then repeated in the discussion. I would recommend picking a format for your results presentation – text or table (does not have to be the same for each group) and reworking the discussion to talk about the importance of the findings, not just what they were.

Simple Summary, Abstract

Line 14: The overall prevalence of 579 samples isn’t based on the population. Repeated samples are not independent.

Line 29: This holds for severe hyperglycemia as well. If I’m reading this correctly, you only have 28 independent samples (hedgehogs) with hyperglycemia, so 39 of the samples are repeats.

Introduction

Lines 43-49: This reads a little awkwardly. Multifactorial and a variety of factors are saying the same thing, so it might read better if this was combined into one thought. I’m not sure about including the stress hyperglycemia information since that’s not really the focus. By being singled out, it sounds like it’s a more important part of your retrospective review, and it’s not addressed. If you were running this as a longitudinal study, it would be good to include subjective information about the animals’ demeanors.

Lines 50-60: You write there’s a negative association between hyperglycemia and disease, implying that the worse the disease, the lower the level of hyperglycemia. All the examples in this paragraph are saying the opposite.

Line 68-69: The prevalence of hyperglycemia in hedgehogs would be using the 252 individual animals, not the 579 blood samples.

Materials and Methods

Lines 79-88: This could be shortened down to the inclusion criteria (had a BG available), any exclusion criteria, and then the definitions of the four groups. I know that you can infer that the fourth group will be normoglycemic, but it would be better to state this clearly, as it is in lines 86-88.

Lines 190-194: Healthy does not equal normoglycemic. You can’t compare the glucose levels if you are using that as a qualifier. The comparison would either be healthy vs sick animals or normoglycemic vs severe hyperglycemic (and then looking at clinical parameters). What happened with the low glycemic and mild hyperglycemic groups? Are there differences in the demographics in these groups?

Line 94: Why were physical disabilities or deformities excluded? If they were considered non-healthy, it should be stated. If they were completely excluded, this should go in your inclusion/exclusion criteria.

Lines 96-97: Did these animals remain healthy for every examination? If that’s true, you might add that for animals that remained healthy during the study period, you used only the first result.

Line 104-105: You should probably state what the factors were.

Lines 108-110: Could isoflurane affect the glucose levels? This is retrospective so this was beyond your control, but it might be a limitation.

Lines 120-123: This says that the F-F and F-F-H test were both run in Python and then the F-F-H test was also run in R. Is that correct? It would be ideal to include the package(s) in R that you used.

Results

Lines 126-127: Were any excluded for physical disabilities or deformities?

Line 128: “The 579 blood glucose test results were classified…” This is explained in the methods. It doesn’t need to be repeated here.

Lines 129-134 and Table 1: The prevalence is based on a population, which in this study is 252 hedgehogs.

Lines 138-143: You have the data in the table, so I don’t know if it needs restating. What would be interesting to know is how this population compares with expected, i.e. How long do pet hedgehogs usually live, what’s a normal weight for an adult? Is 4.4 months adult or are some of these considered juveniles?

Table 2 and Table 3: The sex category should be fixed so that the numbers are MI(MN) and FI(FN), rather than double counting the neutered animals.

Lines 148-149: You have your 28 severe hyperglycemic hedgehogs represented in the table. That makes sense, but I’m not understanding why there’s a sentence about the 579 samples? I am assuming all descriptive data is based on 28 animals and not 67 samples?

Line 162: I would argue that those are neurologic findings, not musculoskeletal.

Lines 158-165: Rather than just stating the data that’s in the table, you might include how many hedgehogs had 2 or 3 or more clinical signs.

Lines 183-184: This categorization should be defined and moved to methods.

Line 193: Table 7 doesn’t reflect the days to recovery.

Lines 193-195: This is methods, and you already have it in 2.3.

Did you look at similarity of signalment in the healthy vs severe hyperglycemic hedgehogs?

Discussion

Lines 206-208: Same issue as above. Also, this is just restating your results. What makes it an important finding?

Lines 218-220: Do you have data to back that up? What’s the baseline to say there’s an increase? With this data, it’s conjecture. The part about cats is interesting, but not relevant to this study.

Lines 237-243: You’re just restating results here. What do those results tell you?

Throughout: Every paragraph restates results in the first sentence.

Lines 247-248: Those are neurologic signs, not lameness.

Lines 250-252: That’s a really sweeping statement that isn’t really supported by your conclusions. Have all other causes of hyperglycemia ruled out, e.g., by necropsy? Can you biologically relate those clinical findings to hyperglycemia?

Lines 259-262: This is the main point in this paragraph. Can you expand on that, rather than just restating your results?

Lines 266-268: Same as above. How many days passed between the original test and the subsequent test? That might be interesting to include in your results.

The conclusion paragraph is a nice summary of the most important findings.

Author Response

For research article

Response to Reviewer X Comments

1. Summary

We would like to sincerely thank the reviewer for the time and effort dedicated to evaluating our manuscript. We greatly appreciate the thoughtful, constructive, and detailed comments, which have significantly contributed to improving the clarity, accuracy, and scientific quality of our work. In response, we have revised the manuscript thoroughly and addressed each point raised by the reviewer. All corresponding changes have been clearly indicated using track changes in the resubmitted version of the manuscript. Below, we provide a point-by-point response to each comment.

2. Questionsfor General Evaluation

Reviewer’s Evaluation

Response and Revisions

Does the introduction provide sufficient background and include all relevant references?

Yes/Can be improved/Must be improved/Not applicable

Are all the cited references relevant to the research?

Yes/Can be improved/Must be improved/Not applicable

Is the research design appropriate?

Yes/Can be improved/Must be improved/Not applicable

Are the methods adequately described?

Yes/Can be improved/Must be improved/Not applicable

Are the results clearly presented?

Yes/Can be improved/Must be improved/Not applicable

Are the conclusions supported by the results?

Yes/Can be improved/Must be improved/Not applicable

3. Point-by-point response to Comments and Suggestions for Authors

Reviewer  General Comment:

This manuscript describes a retrospective review of glucose levels in healthy and hyperglycemic pet hedgehogs. The information regarding the glucose levels is interesting, and publishing this would help exotic/pocket pet veterinarians particularly with the possibility of prognostic markers. However, the comparison of healthy vs severely hyperglycemic animals confuses me. Healthy vs. sick, with rates of glucose categories makes sense, or an exploration of the differences in normo- vs hyperglycemic (e.g. X number of hedgehogs were healthy with hyperglycemia). Additionally, results are presented in the text of the results section, repeated in the tables, and then repeated in the discussion. I would recommend picking a format for your results presentation – text or table (does not have to be the same for each group) and reworking the discussion to talk about the importance of the findings, not just what they were.

General Comment Response:
We sincerely thank the reviewer for the thoughtful and constructive general comments. We appreciate the recognition of the clinical relevance of our findings, and we agree that the structure of group comparisons and the format of results presentation required greater clarity.

In response to your suggestions, we have:

  • Clarified that the healthy and severe hyperglycemia groups were not intended for direct statistical comparison, but rather described independently for contextual understanding (as noted in the revised Introduction, Methods, and Results).
  • Reorganized the Results section to reduce redundancy by streamlining overlapping content between text and tables.
  • Revised the Discussion section to focus more on the interpretation and significance of findings rather than restating numerical results.

These changes were made throughout the revised manuscript and are intended to improve the logical flow, reduce confusion, and enhance clinical relevance. We are grateful for your insights, which helped us substantially improve the structure and clarity of the paper.

Reviewer Comment 1:

(Simple Summary, Abstract) 

Line 14: The overall prevalence of 579 samples isn’t based on the population. Repeated samples are not independent.

Line 29: This holds for severe hyperglycemia as well. If I’m reading this correctly, you only have 28 independent samples (hedgehogs) with hyperglycemia, so 39 of the samples are repeats.

Author Response 1:
We sincerely thank the reviewer for this thoughtful and important observation.

The reported value of 48.1% was calculated based on the total number of blood glucose tests (n = 579), rather than the number of individual animals. We fully acknowledge that repeated measurements from the same individual are not statistically independent, and therefore, this figure does not represent a true prevalence in the population.

The intention behind presenting this value was to address the first objective of the study: to describe the overall distribution of blood glucose measurements across categories (hypoglycemia, normoglycemia, mild hyperglycemia, and severe hyperglycemia). As such, we agree that using the term “prevalence” may be misleading in this context, and we have revised the manuscript to instead use “frequency of hyperglycemic samples” when referring to this result.

Upon reviewing the longitudinal data, we observed that individual hedgehogs could experience a wide range of glycemic states across different visits. In fact, some animals demonstrated normoglycemia, mild hyperglycemia, and severe hyperglycemia at different time points during the three-year study period. This variation made it challenging to assign a stable glycemic status to each individual and calculate prevalence by glycemic category without risking misinterpretation or statistical bias.

For this reason, we only calculated individual-animal-based prevalence for the severe hyperglycemia group. Any hedgehog that demonstrated severe hyperglycemia at least once during the study period was included in this group (n = 28), and prevalence was calculated relative to the total number of animals evaluated (28/252; 11.1%). To avoid pseudoreplication, only the initial blood glucose test showing severe hyperglycemia for each of these individuals was included in downstream statistical analyses and outcome comparisons.

Furthermore, the manuscript and corresponding data tables have been revised to clearly distinguish between sample-based frequencies and individual-based prevalence figures.

(Lines 14-17), (Lines 28-31), (Lines 77-82), (Lines 158-159), (Lines 265-289), (Lines 453-455)

We believe this clarification enhances the transparency and interpretability of our findings. Once again, we thank the reviewer for this valuable feedback, which helped us refine the accuracy of our terminology and analysis presentation.

Reviewer Comment 2 (Lines 43–49):
This reads a little awkwardly. Multifactorial and a variety of factors are saying the same thing, so it might read better if this was combined into one thought. I’m not sure about including the stress hyperglycemia information since that’s not really the focus. By being singled out, it sounds like it’s a more important part of your retrospective review, and it’s not addressed. If you were running this as a longitudinal study, it would be good to include subjective information about the animals’ demeanors.

Author Response 2:
We sincerely thank the reviewer for the thoughtful and constructive feedback.

We agree with your observation regarding the redundancy of the terms “multifactorial” and “a variety of factors.” To improve clarity and conciseness, we have revised the sentence to consolidate these expressions into a single, streamlined statement.

Regarding the mention of stress hyperglycemia, we appreciate your concern about its relevance and placement. However, we respectfully believe that the concept of stress-induced hyperglycemia remains relevant to our study, as hyperglycemia in some hedgehogs may have resulted from physiological stress due to illness, fear, or handling. Stress hyperglycemia is a well-documented phenomenon not only in humans but also in various veterinary species, and it may help explain some of the hyperglycemic episodes observed in our retrospective data. (Lines 53-59)

To reflect this more appropriately, we revised the sentence and added the following line to emphasize the clinical relevance without overemphasizing its role in our study:

“In veterinary medicine, stress hyperglycemia is not only indicative of elevated blood glucose levels but is also utilized as a prognostic factor for disease outcomes.”

Thank you again for your valuable comments, which improved the clarity and focus of our manuscript.

Reviewer Comment 3 (Lines 50–60):
You write there’s a negative association between hyperglycemia and disease, implying that the worse the disease, the lower the level of hyperglycemia. All the examples in this paragraph are saying the opposite.

Author Response 3:
We sincerely thank the reviewer for pointing out this important inconsistency in our original wording.

As the reviewer correctly noted, the phrase "a negative association between hyperglycemia and disease" may be misinterpreted to mean that more severe disease correlates with lower glucose levels, which is the opposite of what the cited studies demonstrate.

To clarify our intended meaning and ensure consistency with the referenced literature, we have revised the text to more accurately reflect a positive association between hyperglycemia and disease severity. The revised sentence now reads: (Lines 58-59)

“Many studies have suggested that hyperglycemia is associated with disease severity and poor prognosis in animals with various diseases [1, 13, 16, 17, 18, 19, 20, 21, 22].”

This revision better conveys that increased blood glucose levels are often linked to more severe illness and worse outcomes, in line with the evidence presented in the referenced studies.

We appreciate the reviewer’s insight, which helped improve the clarity and accuracy of this section.

Reviewer Comment 4 (Lines 68–69):

The prevalence of hyperglycemia in hedgehogs would be using the 252 individual animals, not the 579 blood samples.

Author Response 4:

Thank you for this helpful clarification.

We fully agree with the reviewer’s observation that the term prevalence should be applied to individual animals, not blood samples. In response, we have revised the relevant sentence to distinguish between the two levels of analysis: frequency based on blood samples and prevalence based on individual animals.

The revised sentence now reads: (Lines 77-82)

“The first objective of this study is to investigate the overall frequency of hyperglycemia in blood samples collected from hedgehogs.”

This revision ensures conceptual clarity and consistency with epidemiological terminology. We appreciate the reviewer’s valuable input.

Reviewer Comment 5 (Lines 79–88):
This could be shortened down to the inclusion criteria (had a BG available), any exclusion criteria, and then the definitions of the four groups. I know that you can infer that the fourth group will be normoglycemic, but it would be better to state this clearly, as it is in lines 86–88.

Author Response 5:
We appreciate the reviewer’s helpful suggestion regarding conciseness and clarity.

As pointed out, the original version of this paragraph was overly descriptive and lacked an explicit statement about the normoglycemia group in the initial classification.

In response, we have revised the text to be more concise and to clearly define all four blood glucose groups, including the normoglycemia group, as follows: (Lines 90-96)

“To evaluate the frequency and distribution of hyperglycemia, all blood samples with recorded blood glucose values were included in the analysis. Based on a previously established reference range (89 ± 30 mg/dL) [28], all blood samples were categorized into four groups according to their glucose levels: a hypoglycemia group (<59 mg/dL), a normoglycemia group (59–119 mg/dL), a mild hyperglycemia group (120–179 mg/dL), and a severe hyperglycemia group (≥180 mg/dL).”

This revision improves readability, defines all groups explicitly, and incorporates both inclusion and exclusion criteria as recommended. Thank you again for your valuable feedback.

Reviewer Comment 6 (Lines 190–194):
Healthy does not equal normoglycemic. You can’t compare the glucose levels if you are using that as a qualifier. The comparison would either be healthy vs sick animals or normoglycemic vs severe hyperglycemic (and then looking at clinical parameters). What happened with the low glycemic and mild hyperglycemic groups? Are there differences in the demographics in these groups?

Author Response 6:

(We confirm that our response refers to the content in lines 90–94, not lines 190–194.)

We fully agree with the reviewer’s comment that “Healthy does not equal normoglycemic.” This important clarification highlights the inappropriate comparison between the normoglycemia group in Section 2.1 and the Healthy Hedgehog group in Section 2.2. Your insight allowed us to recognize that our previous classification scheme may cause confusion for readers.

To address this, we revised the subsection title from “2.1. Inclusion Criteria and Classification of Animals” to “2.1. Classification Criteria for All Blood Samples.” This section now focuses solely on the categorization of blood samples. (Lines 90-96)

We would also like to clarify the rationale behind our study design. This study is structured around three independent datasets, each developed with distinct objectives and inclusion criteria:

2.1. Classification Criteria for All Blood Samples – Designed to investigate the frequency of hyperglycemia across all blood samples

2.2. Healthy Hedgehog Criteria – A group of 80 healthy hedgehogs selected for comparison with previously reported reference values for blood glucose.

2.3. Severe Hyperglycemic Hedgehog Criteria – A group of 28 hedgehogs with blood glucose levels ≥180 mg/dL

These three datasets are completely independent and are not intended for direct comparison.

Accordingly, the term “normoglycemia” in Section 2.1 is used solely as a statistical classification based on glucose levels, and it is conceptually distinct from the healthy group defined in Section 2.2.

To reinforce this distinction, we have added the following statement at the beginning of the Results section: (Lines 150-153)

“To improve clarity, we note that the results are structured around three separate datasets, each with unique inclusion criteria and analytical purposes. As such, they represent independent study groups and are not intended for direct intergroup comparison.”

Additionally, in Section 2.2, we replaced the term “normal glucose range” with “reference range” when referring to the previously reported glucose value (89 ± 30 mg/dL). This change ensures conceptual accuracy and prevents the misunderstanding that this range was used as a selection criterion for the healthy group. (Lines 98-99)

We sincerely thank the reviewer again for the insightful suggestions, which helped improve the overall clarity and precision of our manuscript.

Reviewer Comment 7 (Line 94):
Why were physical disabilities or deformities excluded? If they were considered non-healthy, it should be stated. If they were completely excluded, this should go in your inclusion/exclusion criteria.

Author Response 7:
Thank you for your valuable comment, and we apologize for any confusion caused by the previous explanation in Section 2.1.

In this study, the “healthy hedgehog” group described in Section 2.2 was defined independently of the glucose-based classification presented in Section 2.1.
Hedgehogs in the healthy group were those that visited the clinic for routine wellness examinations and showed normal appetite, activity, and physical examination findings. Individuals with physical disabilities or deformities were considered clinically non-healthy and were excluded according to predefined health criteria.

We sincerely thank the reviewer again for the insightful suggestions, which helped improve the overall clarity and precision of our manuscript.

Reviewer Comment 8 (Lines 96–97):
Did these animals remain healthy for every examination? If that’s true, you might add that for animals that remained healthy during the study period, you used only the first result.

Author Response 8:
Thank you for this insightful comment.

As stated in Section 2.2, the healthy hedgehog group was defined independently of the blood glucose-based classification described in Section 2.1. Only animals that were clinically assessed as healthy on the day of the wellness examination were included in this group.

Whether the animals remained clinically healthy throughout the entire study period was not evaluated. Given the relatively short lifespan of hedgehogs, it is unlikely that an individual remained consistently healthy over the three-year data collection period. Therefore, in accordance with the defined criteria, data from only a single time point—the earliest instance at which the animal was both young and clinically healthy—were analyzed for inclusion in the healthy group. A blood glucose value had to be available at that specific time point to qualify for inclusion.

In cases where hedgehogs met the health criteria across multiple wellness examinations, only the data from the first such examination were included in the analysis. This clarification has been explicitly stated in Section 2.2. (Lines 97-106)

We sincerely appreciate the reviewer’s thoughtful suggestion, which helped us clarify the rationale for data selection in the healthy hedgehog group.

Reviewer Comment 9 (Lines 104-105):

You should probably state what the factors were.

Author Response 9:

Thank you for your helpful suggestion.

In response, we have revised the Materials and Methods section to clearly specify the prognostic factors assessed. Specifically, we now state that the evaluated variables included the appetite status on the first day that severe hyperglycemia was confirmed, as well as recovery from severe hyperglycemia.

 We also added definitions for the three appetite-based categories—Good appetite, Decreased appetite, and Complete anorexia—to clarify the classification criteria. (Lines 107-123)

Thank you for your valuable comment, which helped improve the clarity of our study design.

Reviewer Comment 10 (Lines 108–110):

Could isoflurane affect the glucose levels? This is retrospective so this was beyond your control, but it might be a limitation.

Author Response 10:

Thank you for your insightful comment regarding the potential influence of isoflurane on blood glucose levels. We agree that this is an important consideration and appreciate your suggestion.

All clinical procedures in our hospital follow a standardized protocol, and blood collection was always performed immediately after the induction of anesthesia. This method is useful because it allows the blood test results to be available at approximately the same time as those from physical examination, radiography, and ultrasonography, thereby improving diagnostic efficiency. All hedgehogs were initially sedated in an induction chamber using 5% isoflurane in an oxygen flow of 5 L/min for no more than 2 minutes, followed by maintenance anesthesia with 3% isoflurane in an oxygen flow of 1 L/min. Blood sampling was performed first, immediately after sedation, and was completed within 5 minutes from the start of anesthesia in all cases.

To ensure transparency regarding the degree of isoflurane exposure during blood sampling, we have added this detailed information to the Materials and Methods section. (Lines 125-134)

Although several studies in other animal species have shown that isoflurane can induce hyperglycemia, the timing of this effect varies considerably between species and experimental settings. In our study, the duration of isoflurane exposure before blood sampling was brief (less than 5 minutes), reducing the likelihood of significant time-dependent metabolic effects. Moreover, rapid induction helped minimize movement-related stress during the procedure, thereby reducing the possibility of stress-induced hyperglycemia.

To date, no published studies have directly evaluated the effect of isoflurane on blood glucose in hedgehogs. While we believe that the influence of isoflurane is likely minimal in our setting, we acknowledge that this interpretation remains speculative. If isoflurane does indeed cause a rapid metabolic response in hedgehogs, even short exposure might be relevant. Accordingly, we have revised the Materials and Methods section to provide more detailed information on the anesthesia protocol and updated the Conclusion section to note the potential influence of isoflurane on blood glucose, along with other possible contributing factors to hyperglycemia. (Lines 125-134), (Lines 301-305)

Thank you again for your thoughtful suggestion, which has helped to improve the transparency and scientific rigor of our manuscript.

Reviewer Comment 11 (Lines 120–123):
This says that the F-F and F-F-H test were both run in Python and then the F-F-H test was also run in R. Is that correct? It would be ideal to include the package(s) in R that you used.

Author Response 11:
Thank you for your comment. We have clarified the statistical methods used in the revised manuscript.

First, to enhance the analysis of continuous variables such as blood glucose levels, we included independent t-tests to compare two groups (e.g., survivors vs. non-survivors). These tests were performed using the SciPy v1.14.1 library in Python v3.11.5. This addition strengthens the statistical validity of group comparisons for continuous data.

Second, for categorical variables, Fisher–Freeman exact tests were primarily performed using SciPy. However, for contingency tables with more than three categories, Monte Carlo approximated p-values were computed using the base fisher.test function in R v4.3.0 with the simulate.p.value = TRUE option enabled, due to limitations in Python’s support for larger tables. No additional R packages were used beyond base R functionality.

Accordingly, the revised text reads: (Lines 141-148)

“The means, standard deviations (SDs), medians, and confidence intervals (CIs) were calculated for each group. Independent t-tests were used to compare continuous variables between two groups (e.g., survivors vs. non-survivors), and were conducted using SciPy v1.14.1 in Python v3.11.5. Categorical variables were analyzed using Fisher–Freeman exact tests. These were primarily performed using SciPy, but for contingency tables with more than three categories, p-values were additionally computed using a Monte Carlo simulation-based test in R v4.3.0 due to limitations in Python's support for larger tables. A value of p < 0.05 was considered statistically significant.”

We appreciate the reviewer’s suggestion, which allowed us to improve the clarity and completeness of the statistical methods section.

Reviewer Comment 12 (Lines 126–127):
Were any excluded for physical disabilities or deformities?

Author Response 12:
Thank you for your question.

Since Section 3.1, Blood Glucose Level Classification, of the Results examines the distribution of blood glucose values across all 597 blood samples, no animals were excluded based on physical disabilities or deformities.

However, to compare our findings with previously reported reference values((89 ± 30 mg/dL), we separately selected a “healthy hedgehog group.” Only hedgehogs that visited for routine health check-ups, showed normal appetite and activity, and had no abnormalities on physical examination or recent medical treatment were included. Animals with physical disabilities or deformities were not included in this healthy group. The results pertaining to this group are reported separately in Section 3.2 of the Results. The selection criteria for this group are detailed in Section 2.2 (Healthy Hedgehog Criteria), and this group was analyzed independently from the overall dataset described in Section 2.1.

To address this and improve clarity, we have added the following statement at the beginning of the Results section: (Lines 150-153)

“To improve clarity, we note that the results are structured around three separate datasets, each with unique inclusion criteria and analytical purposes. As such, they represent independent study groups and are not intended for direct intergroup comparison.”

We hope that this addition will help readers clearly understand the structure and scope of each dataset and interpret the results accordingly within their respective clinical contexts.

Reviewer Comment 13 (Line 128):
“The 579 blood glucose test results were classified…” This is explained in the methods. It doesn’t need to be repeated here.

Author Response 13:
Thank you for your helpful comment.

We agree that the classification criteria for blood glucose levels were already described in detail in the Methods section (Section 2.1), and repeating them in the Results section is unnecessary.

To avoid redundancy, we have revised the sentence to focus directly on the outcome data, as follows: (Lines 155-158)

“Among the 579 blood glucose test results, 6 were in the hypoglycemia group (<59 mg/dL), 295 in the normoglycemia group (59–119 mg/dL), 211 in the mild hyperglycemia group (120–179 mg/dL), and 67 in the severe hyperglycemia group (≥180 mg/dL), as shown in Table 1.”

We appreciate the reviewer’s suggestion, which helped us improve the clarity and conciseness of the Results section.

Reviewer Comment 14 (Lines 129–134 and Table 1):
The prevalence is based on a population, which in this study is 252 hedgehogs.

Author Response 14:
Thank you for this insightful comment.

We agree that the term “prevalence” should be used with reference to a defined population—in this case, the 252 individual hedgehogs included in the study. In the original sentence, the term was used to describe the proportion of hyperglycemic results among the 579 blood glucose tests, which is more appropriately expressed as a frequency, not a prevalence.

Accordingly, we have revised the sentence to reflect this distinction and avoid confusion in terminology. The updated text now reads: (Lines 158-160)

“The overall frequency of hyperglycemia (≥120 mg/dL) among all samples was 48.1% (278/579). The mean blood glucose level across all samples was 137.8 ± 73.7 mg/dL (range, 28–586).”

Additionally, we have ensured that the term “prevalence” is reserved for describing the number of affected individual animals (e.g., 28/252 hedgehogs with severe hyperglycemia), in accordance with correct epidemiological usage.

We appreciate the reviewer’s comment, which helped us refine the accuracy and clarity of the manuscript.

Reviewer Comment 15(Lines 138–143):
You have the data in the table, so I don’t know if it needs restating. What would be interesting to know is how this population compares with expected, i.e. How long do pet hedgehogs usually live, what’s a normal weight for an adult? Is 4.4 months adult or are some of these considered juveniles?

Author Response 15:
Thank you for this thoughtful comment.

In response, we would like to clarify the nature and purpose of the healthy hedgehog group described in Section 3.2.

The animals in this group were selected based on clinical health status at the time of presentation for routine wellness examinations. Only those with normal appetite, activity levels, and physical examination findings were included. As a result, this group includes relatively younger animals, with a mean age of 17.4 months. Upon review of the raw data, we found that 8 of the 80 hedgehogs in this group were younger than 6 months, with the youngest being 4.4 months old.

Regarding the classification of age, published sources indicate that African pygmy hedgehogs typically reach sexual maturity around 2 to 3 months of age. For example, the Merck Veterinary Manual states that sexual maturity occurs at 2–3 months [Merck Veterinary Manual: Overview of Hedgehogs.https://www.merckvetmanual.com/exotic-and-laboratory-animals/hedgehogs/overview-of-hedgehogs],

and the AnAge database reports a mean age of sexual maturity of approximately 84 days (2.8 months). [AnAge Database. Atelerix albiventris. https://genomics.senescence.info/species/entry.php?species=Atelerix_albiventris].

Mitchell and Tully (2009) report male sexual maturity at 2–6 months and female sexual maturity at 6–8 months [Mitchell, M.A.; Tully, T.N., Jr. Hedgehogs. In Manual of Exotic Pet Practice; Mitchell, M.A., Tully, T.N., Jr., Eds.; Saunders Elsevier: St. Louis, MO, USA, 2009; pp. 401–415].

Thus, from a biological standpoint, a 4.4-month-old hedgehog may be considered an early-stage adult. However, in clinical practice, we often regard animals under 6 months of age as juvenile-like, even if they are sexually mature. Some individuals even reach their adult body size by around 3 months and maintain it throughout life.

As for normal adult body weight, the Merck Veterinary Manual lists a typical adult weight range of 250–600 g, while the AnAge database notes an average of 500 g. Mitchell and Tully report adult male weights of 500–600 g and adult female weights of 250–400 g.

That said, our healthy hedgehog group was not designed to represent the general healthy population. Instead, it was selectively defined to serve a specific analytical purpose—namely, to compare the blood glucose levels of clinically healthy hedgehogs in this study with previously reported reference values. Given the selective inclusion criteria and age distribution, it would be inappropriate to generalize this group to represent all healthy adult hedgehogs.

We have clarified this point in the revised manuscript. Thank you again for your helpful question, which allowed us to more clearly describe the scope and limitations of this subgroup.

Reviewer Comment 16 (Table 2 and Table 3):
The sex category should be fixed so that the numbers are MI(MN) and FI(FN), rather than double counting the neutered animals.

Author Response 16:
Thank you for this valuable comment.

Upon review, we confirmed that the original formatting of sex categories in Tables 2 and 3 had listed neutered animals as separate counts, potentially resulting in double counting within each sex group.

To address this, we have revised the sex categorization in both tables to clearly distinguish between intact and neutered individuals using the following abbreviation system: (Tables 2, Tables 3)

IM = Intact Male

NM = Neutered Male

IF = Intact Female

SF = Spayed Female

All individuals are now counted only once, under their respective and mutually exclusive category. We have also added a note below each table to define these abbreviations and ensure clarity for the reader.

We appreciate the reviewer’s suggestion, which has contributed to improving the accuracy, clarity, and consistency of the manuscript.

Reviewer Comment 17 (Lines 148–149):
You have your 28 severe hyperglycemic hedgehogs represented in the table. That makes sense, but I’m not understanding why there’s a sentence about the 579 samples? I am assuming all descriptive data is based on 28 animals and not 67 samples?

Author Response 17:
Thank you for your insightful comment.

As you correctly noted, the results presented in Section 3.3 (Severe Hyperglycemia Hedgehog Group) are based on the data from the first day on which severe hyperglycemia was identified in each of the 28 individual hedgehogs. Therefore, all descriptive statistics in this section are based on these 28 animals, not on the 67 blood samples classified as severe hyperglycemia.

In the original version of the manuscript, we included the number of severe hyperglycemia samples (67/579; 11.6%) to highlight the similarity between the frequency of sample-based severe hyperglycemia and the prevalence based on individual hedgehogs (28/252; 11.1%). However, as you rightly pointed out, this reference could create confusion by conflating sample-level and animal-level metrics within the same context.

Accordingly, we have removed the reference to the 579 blood samples from Section 3.3 to maintain clarity and consistency in the presentation of the data. (Lines 176-177)

We sincerely appreciate your comment, which helped us improve the clarity of the Results section.

Reviewer Comment 18 (Line 162):

Line 162: I would argue that those are neurologic findings, not musculoskeletal.

Author Response 18:

Thank you for your comment.

We completely agree with your assessment that the gait abnormalities described in our study are more likely to be neurologic in origin rather than musculoskeletal.

These clinical signs—such as limb paresis, head tilt, and turning—are highly suggestive of either central or peripheral nervous system dysfunction and may be closely associated with conditions such as wobbly hedgehog syndrome (WHS).

Although we initially described these findings as "lameness presumed to be neurologic" to reflect diagnostic uncertainty, we now recognize that this wording may still lead to confusion. To improve accuracy and clarity, we have revised the terminology throughout the manuscript and in Table 4. Specifically, we replaced “lameness” with “neurologic signs,” as this term more precisely reflects the nature of the clinical findings. (Lines 192-195), (Table 4), (Lines 330-333)

We appreciate your feedback, which helped us improve the precision and consistency of our terminology.

Reviewer Comment 19 (Lines 158-165):

Lines 158-165: Rather than just stating the data that’s in the table, you might include how many hedgehogs had 2 or 3 or more clinical signs

Author Response 19:
Thank you for your constructive comment.

In response, we have revised the text in the Results section to include a detailed breakdown of the number of hedgehogs that exhibited multiple concurrent clinical signs. Specifically, we now state how many hedgehogs exhibited 0 to 6 clinical signs, as well as the number of hedgehogs with ≥2, ≥3, ≥4, ≥5, and ≥6 signs. This provides a clearer picture of the overall clinical burden in the severe hyperglycemia group. Additionally, we have included the most common combinations of symptoms observed among these animals. The revised paragraph is as follows: (Lines 198-208)

“Many of the hedgehogs exhibited multiple concurrent clinical signs. Among the 28 hedgehogs with severe hyperglycemia, only one was asymptomatic and one exhibited a single clinical sign. In total, 26 out of 28 hedgehogs (92.9%) had two or more clinical signs, 25 (89.3%) had three or more, 20 (71.4%) had four or more, 13 (46.4%) had five or more, and 4 (14.3%) had six or more. No hedgehogs exhibited more than six clinical signs. Among the 26 hedgehogs with two or more signs, the most common combination was anorexia + loss of activity, observed in 21 hedgehogs. The second most frequent combinations were anorexia + abnormal feces and loss of activity + abnormal feces, each identified in 16 hedgehogs. In the 25 hedgehogs with three or more clinical signs, the most common triad was anorexia + loss of activity + abnormal feces, noted in 15 hedgehogs.”

We hope this addition sufficiently addresses your recommendation and enhances the clarity and comprehensiveness of the clinical profile data.

Reviewer Comment 20 (Lines 183-184):

Lines 183-184: This categorization should be defined and moved to methods.

Author Response 20:

Thank you for your insightful comment.

In response, we have added a detailed definition of the appetite-based categorization to the Materials and Methods section. Specifically, hedgehogs were categorized into three groups based on the degree of appetite loss observed on the first day severe hyperglycemia was confirmed: the Good appetite group (normal food intake), the Decreased appetite group (noticeable reduction in food intake but continued eating), and the Complete anorexia group (no food intake observed for more than 24 hours). (Lines 113-123)

Accordingly, we revised the corresponding text in the Results section to reference this classification, now defined in the Methods, and to focus solely on the associated mortality rates within each group (Table 6). (Lines 227-233)

This revision improves both clarity and structural consistency across the manuscript.

Reviewer Comment 21 (Line 193):
Table 7 doesn’t reflect the days to recovery.

Author Response 21:
Thank you for your valuable comment.

Upon review, we found that the sentence “The average time to recovery from hyperglycemia was 14.4 days (range, 7–30 days)” was appropriately included earlier in the paragraph associated with Table 5. However, due to an editorial oversight, the same sentence was inadvertently repeated in the paragraph introducing Table 7. We sincerely apologize for this redundancy.

To resolve this, we have removed the duplicated sentence from the Table 7 section. This paragraph now focuses solely on the comparison of 100-day mortality rates based on recovery status from severe hyperglycemia.

We appreciate your comment, which allowed us to improve the clarity and consistency of the manuscript.

Reviewer Comment 22 (Lines 193–195):

This is methods, and you already have it in 2.3.

Author Response 22:

Thank you for your valuable feedback.

 In response to your comment, we have revised this section to focus solely on the results rather than repeating methodological details already described in Section 2.3. Specifically, we removed the procedural description ("Twenty-eight hedgehogs were classified into two groups...") and rephrased the paragraph to highlight the outcome of the analysis. The revised text now reads: (Lines 236-242)

“Among the 28 hedgehogs, follow-up blood glucose tests were conducted an average of 14.6 days after the initial diagnosis (range, 3–30 days), during which 12 animals were confirmed to have recovered from severe hyperglycemia. The average time to confirmed recovery among these 12 hedgehogs was 14.4 days (range, 7–30 days). The mortality rate was 8.3% (1/12) in the recovery group and 87.5% (14/16) in the non-recovery group (Table 7). Hedgehogs that had recovered from severe hyperglycemia by the time of follow-up testing had a significantly lower risk of death within 100 days (p < 0.001).”

This revision ensures that the Results section presents only the findings, while the classification criteria and analysis methods remain clearly defined in the Methods section (2.3).

Thank you sincerely for your comment, which helped improve the clarity of our results presentation and the structural consistency of the manuscript.

Reviewer Comment 23 (Lines 193–195):
Did you look at similarity of signalment in the healthy vs severe hyperglycemic hedgehogs?

Author Response 23:
Thank you for this meaningful question.

In the early stages of analysis, we did consider comparing signalment (age, body weight, blood glucose level, and sex) between the 28 hedgehogs with severe hyperglycemia and the 80 hedgehogs classified as healthy. However, we ultimately determined that the healthy group was not suitable for statistical comparison. These hedgehogs were not selected as a randomized control group; instead, they were chosen based on specific inclusion criteria. They had visited the hospital for routine health checkups, and when multiple visits occurred, only data from the first visit were used. As a result, the healthy group tended to be younger and lighter in body weight, which introduced significant selection bias.

Due to this limitation, we concluded that a direct comparison of signalment between the healthy and severely hyperglycemic groups would not be appropriate and was not performed in this study.

However, based on the reviewer’s insightful suggestion, we conducted a comparison of signalment within the severe hyperglycemia group by dividing the 28 hedgehogs into survivor and non-survivor subgroups. We compared blood glucose level (mg/dL), body weight (g), and age (months) at the time of initial diagnosis. The results of this analysis, including statistical significance, have been newly added to the Results section and are presented in Table 8. (Lines 245-252)

This analysis has further strengthened the clinical relevance of our study, and we sincerely thank the reviewer for the valuable suggestion.

Discussion

Reviewer Comment 24 (Lines 206–208):
Same issue as above. Also, this is just restating your results. What makes it an important finding?

Author Response 24:

 Thank you for your valuable comment.

As you pointed out, the original sentence appeared to restate the results without emphasizing its significance. However, our intention was to present the finding at the beginning of the paragraph to facilitate comparison with previously reported hyperglycemia rates in other species.

We agree that the placement of the sentence may have disrupted the logical flow and made it seem redundant.

To improve clarity and coherence, we have moved this sentence to the middle of the paragraph and revised the surrounding text to better highlight the clinical relevance and context of our findings.

The following statement has been added to the Discussion section to highlight the importance of this finding: (Lines 284-289)

“In this study, the high frequency of hyperglycemia among all blood samples (48.1%) suggests that veterinarians may frequently encounter hyperglycemia during the clinical care of hedgehogs. The frequency of severe hyperglycemia was 11.6%, indicating that veterinarians may occasionally encounter severe hyperglycemia during the clinical care of hedgehogs. This rate of occurrence emphasizes the importance of this study on severe hyperglycemia in hedgehogs.”

Your comment helped us emphasize the significance of this result. Thank you.

Reviewer Comment 25 (Lines 218-220):

Do you have data to back that up? What’s the baseline to say there’s an increase? With this data, it’s conjecture. The part about cats is interesting, but not relevant to this study.

Author Response 25:
Thank you for your insightful comment.

As you correctly pointed out, the original sentence lacked a clear baseline and could be interpreted as a speculative statement.

In response, we have removed the vague expression and added supporting data from a small subgroup analysis within our study to better clarify the interpretation.

Specifically, among the 252 hedgehogs included in the study, we identified a subgroup of 20 individuals who underwent five or more blood glucose measurements. A total of 193 samples were obtained from this subgroup, and the frequency of hyperglycemia was 54.9% (106/193), which is higher than the overall frequency of 48.1% (278/579) observed in all samples. This finding suggests that the high frequency of hyperglycemia observed in our study may be partially explained by increased hospital visits and repeated blood sampling in sick animals, which may have led to an overrepresentation of hyperglycemic cases. (Lines 265-280)

Additionally, while the previously cited study on stress-induced hyperglycemia in cats [15] does not directly apply to hedgehogs, we referenced it to illustrate that similar patterns of sampling bias have been observed in other species. The text has been revised to ensure this context is made clearer and more relevant to our findings.

We appreciate your comment, which helped us strengthen the clarity and validity of our interpretation.

Reviewer Comment 26 (Lines 237-243):

You’re just restating results here. What do those results tell you?

Throughout: Every paragraph restates results in the first sentence.

Author Response 26:
Thank you for your helpful feedback.

This section was originally written to follow the structure of presenting the result, interpreting its meaning, and comparing it with previous findings. However, we agree that the repeated use of raw data without adequate contextual explanation may have made the paragraph appear as a restatement of results rather than a meaningful discussion.

In response to your comment, we have revised the paragraph to incorporate the authors’ interpretation of the findings rather than merely repeating the numerical outcomes.

The revised paragraph now provides a more detailed overview of studies in other species where hyperglycemia has been identified as a prognostic indicator. Based on the high mortality rate observed in hedgehogs with severe hyperglycemia, we suggest that severe hyperglycemia may also be associated with poor outcomes in this species. (Lines 360-384)

We are grateful for your comment, which allowed us to refine the scientific context and enhance the overall quality of the discussion.

Reviewer Comment 27 (Lines 247-248):

Those are neurologic signs, not lameness.

Author Response 27:
Thank you for your insightful comment.

We agree with your observation that the term "lameness" is not appropriate for describing the clinical signs observed in our study, as it typically refers to musculoskeletal issues. The signs reported are more consistent with neurologic dysfunction.

Although we initially described these findings as "lameness presumed to be neurologic" to reflect diagnostic uncertainty, we now recognize that this wording may still lead to confusion.

To improve accuracy and clarity, we have revised the terminology throughout the manuscript and in Table 4. Specifically, we replaced “lameness” with “neurologic signs,” as this term more precisely reflects the nature of the clinical findings. (Lines 192-195), (Table 4), (Lines 330-333)

We appreciate your feedback, which helped us improve the precision and consistency of our terminology.

Reviewer Comment 28 (Lines 250-252):

That’s a really sweeping statement that isn’t really supported by your conclusions. Have all other causes of hyperglycemia ruled out, e.g., by necropsy? Can you biologically relate those clinical findings to hyperglycemia?

Author Response 28:
Thank you for your thoughtful comment and for pointing out the need for caution when interpreting the relationship between clinical abnormalities and hyperglycemia.

In response, we have substantially revised the paragraph to avoid making an overly broad conclusion. Instead of stating that physical defects are likely causes of stress hyperglycemia in hedgehogs, we now present relevant evidence from previous studies in other animal species showing that physical abnormalities, pain, and underlying diseases can induce stress-related hyperglycemia. These include hyperglycemia in cattle with mucosal disease or hemorrhagic bowel syndrome, in dogs with fractures, and in animals with head trauma.

We then describe the clinical findings observed in our study—such as tumors and neurologic signs—in a more objective and descriptive manner, suggesting that these factors may contribute to stress-induced hyperglycemia in hedgehogs. However, we clearly acknowledge that hyperglycemia is influenced by multiple factors and that causality cannot be established based on our data. To address this, we added a concluding statement emphasizing the need for further research, including histopathological analysis, to better understand these associations. (Lines 318-336)

We appreciate your comment, which helped us refine the interpretation and improve the scientific accuracy and clarity of the discussion.

Reviewer Comment 29 (Lines 259-262):

This is the main point in this paragraph. Can you expand on that, rather than just restating your results?

Author Response 29:
Thank you for your valuable comment and for highlighting the need to expand on the main point of this paragraph.

In response, we revised the paragraph to go beyond simply restating the results and instead provided a more in-depth interpretation of the clinical relevance of anorexia in relation to prognosis.

We now discuss the biological and clinical context of anorexia as a marker of systemic disease severity, supported by findings from previous studies in other mammals, including rabbits. These studies demonstrate that complete anorexia is often associated with life-threatening conditions and can be used as a prognostic indicator in cases of hyperglycemia.

We also explain how, in clinical settings, appetite is a practical and easily observable parameter that may be especially useful in species such as hedgehogs, where assessing overall clinical status can be challenging. Therefore, evaluating the severity of anorexia at the time of diagnosis may offer clinicians a valuable early tool for predicting outcomes and guiding treatment intensity. (Lines 385-406)

We appreciate your suggestion, which helped us strengthen the depth and clinical applicability of our discussion.

Reviewer Comment 30 (Lines 266–268):
This is the main point in this paragraph. Can you expand on that, rather than just restating your results? How many days passed between the original test and the subsequent test? That might be interesting to include in your results.

Author Response 30:
Thank you for your thoughtful comment.

In response, we revised the paragraph to provide a more in-depth interpretation of the prognostic value of recovery from severe hyperglycemia, rather than simply restating the numerical results. Specifically, we included the mean interval between the initial and follow-up tests (14.6 days) to address your question regarding the timing of reassessment. We also expanded the discussion by noting that although recovery from hyperglycemia may be a helpful prognostic indicator, it should be used cautiously and in conjunction with other clinical signs such as appetite, general condition, and resolution of symptoms.

Furthermore, we acknowledged that the relatively long follow-up intervals in our study limit the ability to determine the optimal timing for prognostic evaluation based on glucose normalization. Despite these limitations, the substantial difference in mortality rates between recovered and non-recovered groups supports the potential clinical utility of this parameter. (Lines 407-430)

We appreciate your suggestion, which helped us strengthen the clinical relevance and contextual depth of this section.

4. Response to Comments on the Quality of English Language

Point 1: The English is fine and does not require any improvement.

Response 1: We sincerely appreciate your positive feedback on the quality of English used in the manuscript.

5. Additional clarifications

We would like to express our sincere gratitude to the reviewer for the thoughtful, detailed, and insightful comments provided throughout the review process. Your feedback has greatly contributed to improving the clarity, rigor, and scientific value of our manuscript.

While we have made every effort to address all comments thoroughly, we would especially appreciate your further assessment regarding one remaining concern—the inclusion of the healthy hedgehog group.

To compare the previously reported reference blood glucose value of 89 ± 30 mg/dL with the mean glucose level observed in our study’s healthy hedgehog group, we selected healthy individuals using a new set of criteria that were independent of those applied to the other groups. Details on this group are provided in the Materials and Methods section (2.2. Healthy Hedgehog Criteria), the Results section (3.2. Healthy Hedgehog Group), and the relevant part of the Discussion.

However, we acknowledge that our original descriptions may have unintentionally suggested that this group was established for direct statistical comparison with the severe hyperglycemia group. To clarify this potential source of confusion, we have added explicit statements in all relevant sections stating that the healthy group was constructed for descriptive purposes only and was not intended as a statistical control. If this clarification is still considered insufficient or potentially misleading, we would be grateful for further guidance. We are fully prepared to remove this subset from the manuscript if necessary, and we believe that the integrity and core findings of the study would remain intact even without it.

In addition, we would like to provide the following clarifications:

  • Terminology Adjustments: In the revised manuscript, we have clarified the use of the terms "frequency" and "prevalence" in accordance with the level of analysis (sample-level vs. individual-level). This distinction was made to ensure terminological accuracy and to better reflect the structure of our dataset.
  • Structure of the Results Section: To avoid misinterpretation, we have added a statement at the beginning of the Results section emphasizing that the findings are based on three independent datasets with distinct inclusion criteria. This clarification aims to prevent readers from assuming statistical comparability between the groups.
  • Line Number Referencing: All major revisions and additions have been highlighted using track changes in the resubmitted Word manuscript. In the resubmitted PDF file, line numbers have been referenced throughout the response document to assist reviewers and editors in locating the changes efficiently.

Once again, we are deeply grateful for your guidance. Your comments helped us identify and revise elements that could have caused confusion for readers, and we truly appreciate your contribution to improving the transparency and scientific quality of our work.

Reviewer 2 Report

Comments and Suggestions for Authors

Please see all my comments within the paper. 

Author Response

For research article

Response to Reviewer X Comments

1. Summary

We would like to sincerely thank the reviewer for the time and effort dedicated to evaluating our manuscript. We greatly appreciate the thoughtful, constructive, and detailed comments, which have significantly contributed to improving the clarity, accuracy, and scientific quality of our work. In response, we have revised the manuscript thoroughly and addressed each point raised by the reviewer. All corresponding changes have been clearly indicated using track changes in the resubmitted version of the manuscript. Below, we provide a point-by-point response to each comment.

2. Questionsfor General Evaluation

Reviewer’s Evaluation

Response and Revisions

Does the introduction provide sufficient background and include all relevant references?

Yes/Can be improved/Must be improved/Not applicable

Are all the cited references relevant to the research?

Yes/Can be improved/Must be improved/Not applicable

Is the research design appropriate?

Yes/Can be improved/Must be improved/Not applicable

Are the methods adequately described?

Yes/Can be improved/Must be improved/Not applicable

Are the results clearly presented?

Yes/Can be improved/Must be improved/Not applicable

Are the conclusions supported by the results?

Yes/Can be improved/Must be improved/Not applicable

3. Point-by-point response to Comments and Suggestions for Authors

Reviewer Comment 1:

Blood glucose levels vary greatly depending on time since eating (fasting duration) and what was fed. Unless fasting duration is known, this study makes little sense. Stress levels have to be standardized including heart rate under isoflurane anesthesia. Length of time under iso before blood was drawn is also important.

Obesity also plays a role. How many of these with "hyperglycemia" were obese? Age is looked at, and sex although neutered vs intact isn't compared. Also, for intact females - estrus stage or reproductive stage?  This might be difficult to tell unless vaginal swab done or behavioral lordosis noted. Were any pregnant?

Please revise and discuss the body score, age, sex, reproductive status (i.e. estrus, pregnant?) and time since fasting plus what was the last food item consumed in order to work with the blood glucose levels.

Serial testing done in an individual timed after eating would yield a better way to establish normal reference ranges.

Author Response 1 (1-1 – 1-5):
We sincerely thank the reviewer for the thoughtful and insightful feedback regarding the relationship between physiological variables and blood glucose levels in this study. The points raised are highly valid, and we have addressed them thoroughly as follows and revised the manuscript accordingly:

1-1. Lack of information on fasting duration and diet

Author Response 1-1:

We sincerely thank the reviewer for highlighting the critical importance of standardizing pre-analytical variables such as fasting duration, timing of blood collection, and the composition of the last meal when interpreting blood glucose values. We fully agree that these factors can significantly affect glucose concentrations, particularly in nocturnal species such as hedgehogs. As the reviewer correctly noted, assuming a fasted state without clear documentation of the time since the last meal or the animal's feeding behavior introduces uncertainty and must be approached with caution.

Due to the retrospective nature of our study, detailed information on fasting duration, exact timing of the last meal, and standardization of blood sampling time was not available in the medical records. This limitation has been explicitly acknowledged in the revised manuscript. (Lines 432-441)

Nonetheless, our approach is consistent with previously published studies that reported blood glucose values in hedgehogs under clinical or rehabilitation settings without strict fasting control. For example, Rosa et al. (2023) reported a reference interval of 108.5 ± 24.1 mg/dL in healthy European hedgehogs (Erinaceus europaeus) without providing information on the animals’ feeding or fasting status, and this study was published in Animals (MDPI).
In another study, Rossi et al. (2014) reported a mean blood glucose level of 106.2 ± 14.4 mg/dL (range, 77.4–131.4 mg/dL) in 30 rehabilitated wild European hedgehogs. Importantly, the authors explicitly noted that fasting prior to blood collection could not be controlled due to the nature of wildlife rehabilitation.

We believe that the publication of these studies—despite lacking standardized fasting protocols—reflects the research community’s recognition of the practical limitations and the urgent need to build baseline physiological data for hedgehogs. Our study shares this aim, and we hope it will contribute to filling this gap by providing preliminary, clinically derived blood glucose data from healthy individuals.

In addition, we have revised the manuscript to avoid suggesting a formal “normal blood glucose range.” Instead, we now refer to “the mean blood glucose level,” which more appropriately reflects the nature of our study design and its inherent limitations.

Once again, we greatly appreciate the reviewer’s insights, which helped refine the clarity, scientific rigor, and transparency of our work.

1-2. Impact of isoflurane anesthesia

Author Response 1-2:

Thank you for your insightful comment regarding the potential influence of isoflurane on blood glucose levels. We agree that this is an important consideration and appreciate your suggestion.

All clinical procedures in our hospital follow a standardized protocol, and blood collection was always performed immediately after the induction of anesthesia. This method is useful because it allows the blood test results to be available at approximately the same time as those from physical examination, radiography, and ultrasonography, thereby improving diagnostic efficiency. All hedgehogs were initially sedated in an induction chamber using 5% isoflurane in an oxygen flow of 5 L/min for no more than 2 minutes, followed by maintenance anesthesia with 3% isoflurane in an oxygen flow of 1 L/min. Blood sampling was performed first, immediately after sedation, and was completed within 5 minutes from the start of anesthesia in all cases.

To ensure transparency regarding the degree of isoflurane exposure during blood sampling, we have added this detailed information to the Materials and Methods section. (Lines 126-130)

Although several studies in other animal species have shown that isoflurane can induce hyperglycemia, the timing of this effect varies considerably between species and experimental settings. In our study, the duration of isoflurane exposure before blood sampling was brief (less than 5 minutes), reducing the likelihood of significant time-dependent metabolic effects. Moreover, rapid induction helped minimize movement-related stress during the procedure, thereby reducing the possibility of stress-induced hyperglycemia.

To date, no published studies have directly evaluated the effect of isoflurane on blood glucose in hedgehogs. While we believe that the influence of isoflurane is likely minimal in our setting, we acknowledge that this interpretation remains speculative. If isoflurane does indeed cause a rapid metabolic response in hedgehogs, even short exposure might be relevant. Accordingly, we have revised the Materials and Methods section to provide more detailed information on the anesthesia protocol and updated the Conclusion section to note the potential influence of isoflurane on blood glucose, along with other possible contributing factors to hyperglycemia. (Lines 126-130), (Lines 301-305)

Thank you again for your thoughtful suggestion, which has helped to improve the transparency and scientific rigor of our manuscript.

1-3. Obesity and body condition score (BCS)

Author Response 1-3:

Hsieh et al. (2015) reported a mean body weight of 401 ± 98 g in 63 African pygmy hedgehogs housed at the Taipei Zoo. This value is similar to the mean body weight of 496.7 ± 146.1 g in the 80 healthy hedgehogs included in our study. In contrast, the mean body weight of the 28 hedgehogs with severe hyperglycemia was 569.3 ± 161.2 g, suggesting a trend toward higher body weight in this group.

However, it is important to note that the healthy hedgehog group in our study was not randomly selected to represent the general healthy population. Rather, this group was defined specifically to allow comparison of blood glucose values with the previously reported reference value of 89 ± 30 mg/dL. The group was selected from medical records of hedgehogs that presented for routine health check-ups, using the earliest available data point with a recorded blood glucose measurement. Notably, this group includes some individuals who later developed severe hyperglycemia, meaning their data were obtained during a period when they were clinically healthy. Therefore, we determined that it would be statistically inappropriate to compare the body weights of the healthy group and the severe hyperglycemia group. Accordingly, the revised manuscript clearly states that these two groups are not comparable.

In response to the reviewer’s suggestion, we have incorporated additional data related to obesity within the severe hyperglycemia group. Specifically, we compared biometric parameters between survivors and non-survivors. The Results section now includes body weight, age, sex, and initial blood glucose levels of hedgehogs that survived versus those that died within 100 days of diagnosis, along with statistical comparisons. (Lines 179-181), (Table 8), (Lines 245-252)

In the Discussion section, we referenced published literature on body weight ranges in hedgehogs and discussed the trend toward obesity observed in our severe hyperglycemia group. We also added a discussion of the potential relationship between body weight and survival outcome in these animals. (Lines 337-358)

1-4. Reproductive status (neutering, estrus, pregnancy)

Author Response 1-4:

We would like to take this opportunity to explain the rationale behind our study design and analytical approach.

This study was based on three independent datasets, each constructed with different objectives and inclusion criteria:

Blood Glucose Frequency Dataset (579 blood samples):

This dataset includes all blood glucose values collected from 252 hedgehogs, including repeated measurements from the same individuals. As the analysis was conducted at the sample level, not the individual level, we have revised the manuscript to replace the term prevalence with frequency. The purpose of this dataset was to descriptively evaluate the overall distribution of glucose concentrations across the entire study population.

Healthy Hedgehog Group (n = 80):

This group was constructed using the earliest available glucose value from hedgehogs that were clinically healthy at the time of routine wellness examinations. Data on age, body weight, blood glucose level, and sex were collected to serve as a reference for descriptive comparison with previously reported normal glucose ranges (e.g., 89 ± 30 mg/dL). As this group was not randomly selected or matched, it was not used as a statistical control group for inferential analysis.

Severe Hyperglycemia Group (n = 28):

This group included hedgehogs with an initial blood glucose concentration of ≥180 mg/dL at the time of presentation. We collected information on age, sex, body weight, glucose level, major clinical signs, and survival outcome to descriptively explore potential prognostic indicators associated with severe hyperglycemia.

Because the three groups differed in terms of purpose, selection strategy, and clinical context, we determined that direct statistical comparisons were not appropriate. Essential assumptions such as randomization, independence, and comparability could not be satisfied. In particular, constructing a valid control group for the severe hyperglycemia cohort would have required matching for variables such as age, time of presentation, and comorbidities—an approach that was not feasible due to the retrospective nature of the dataset and variability in clinical presentation.

In response to the reviewer’s suggestion, we also explored the distribution of sex between the healthy hedgehog group and the severe hyperglycemia group using the chi-square test. The result indicated no statistically significant difference (p = 0.844). (Lines 337-343)

However, as the healthy group was not randomly selected but rather constructed under specific inclusion criteria (i.e., based on the earliest available glucose measurement in clinically healthy individuals), we consider it methodologically inappropriate to treat this group as a valid comparator for inferential analysis. For the same reason, variables such as age and body weight were not compared between the two groups, in order to avoid potential bias introduced by non-random selection. Therefore, although the analysis was conducted, we have decided not to include these comparisons in the manuscript to maintain scientific rigor.

We also recognize that, despite being analyzed independently, the presentation of the healthy and severe hyperglycemia groups in the original manuscript may unintentionally imply comparability. To address this and improve clarity, we have added the following statement at the beginning of the Results section: (Lines 150-153)

“To improve clarity, we note that the results are structured around three separate datasets, each with unique inclusion criteria and analytical purposes. As such, they represent independent study groups and are not intended for direct intergroup comparison.”

We hope that this addition will help readers clearly understand the structure and scope of each dataset and interpret the results accordingly within their respective clinical contexts.

Your comment was highly valuable in helping us refine the clarity and transparency of our manuscript.

1-5. Serial glucose testing in relation to postprandial timing

Author Response 1-5:

We fully agree with the reviewer that a prospective study involving serial glucose measurements at defined time intervals after feeding would be highly valuable for distinguishing between normal and pathological hyperglycemia. We appreciate this insightful suggestion. Although such an approach was not feasible in the present retrospective study, we recognize its importance for future research in this field.

Reviewer Comment 2:

I think you are assuming that these pet hedgehogs had not eaten since maybe the night before - but were all the blood tests run the same number of hours after a meal? Same time of day? Without knowing the length of time from the last meal, how long that fasting period was, and even what the last meal was can alter your results. Hedgies bled early in the morning may have higher levels than ones drawn mid-day when normally they would be asleep. These parameters need to be standardized to establish "normal" and "abnormal" reference ranges.

Author Response 2:

We sincerely thank the reviewer for highlighting the critical importance of standardizing pre-analytical variables such as fasting duration, timing of blood collection, and the composition of the last meal when interpreting blood glucose values. We fully agree that these factors can significantly affect glucose concentrations, particularly in nocturnal species such as hedgehogs. As the reviewer correctly noted, assuming a fasted state without clear documentation of the time since the last meal or the animal's feeding behavior introduces uncertainty and must be approached with caution.

Due to the retrospective nature of our study, detailed information on fasting duration, exact timing of the last meal, and standardization of blood sampling time was not available in the medical records. This limitation has been explicitly acknowledged in the revised manuscript. (Lines 432-441)

Nonetheless, our approach is consistent with previously published studies that reported blood glucose values in hedgehogs under clinical or rehabilitation settings without strict fasting control. For example, Rosa et al. (2023) reported a reference interval of 108.5 ± 24.1 mg/dL in healthy European hedgehogs (Erinaceus europaeus) without providing information on the animals’ feeding or fasting status, and this study was published in Animals (MDPI).

In another study, Rossi et al. (2014) reported a mean blood glucose level of 106.2 ± 14.4 mg/dL (range, 77.4–131.4 mg/dL) in 30 rehabilitated wild European hedgehogs. Importantly, the authors explicitly noted that fasting prior to blood collection could not be controlled due to the nature of wildlife rehabilitation.

We believe that the publication of these studies—despite lacking standardized fasting protocols—reflects the research community’s recognition of the practical limitations and the urgent need to build baseline physiological data for hedgehogs. Our study shares this aim, and we hope it will contribute to filling this gap by providing preliminary, clinically derived blood glucose data from healthy individuals.

In addition, we have revised the manuscript to avoid suggesting a formal “normal blood glucose range.” Instead, we now refer to “the mean blood glucose level,” which more appropriately reflects the nature of our study design and its inherent limitations.

Once again, we greatly appreciate the reviewer’s insights, which helped refine the clarity, scientific rigor, and transparency of our work.

Reviewer Comment 3:

only if completely anorexic? you need to discuss hepatic lipidosis of anorectic hedgehogs and how this impacts blood glucose levels. As omnivore/insectivore/carnivores, hedgehogs that have not eaten for more than 24 hours may begin to develop hepatic lipidosis, and mobilize blood gluocose. An ill one with elevated glucose that you get to eat a high density food (like Critical Care Carnivore) , then the next day you recheck -- did these remain elevated, or did you see a decrease? (i.e. beginning hepatic lipidosis reversal?)  It is critical to note that prognosis depends heavily on getting the hedgehog to ingest food when presented ill. In my experience, if you can reverse the hepatic lipidosis then within a few days the blood glucose decreases and the hedgie will survive. If I based the next day's level as my basis for prognosis, it is not grave as you are indicating, but guarded, and it may take a few days before that parameter can be reversed. I do not disagree that an elevated blood glucose is not a problem - but only giving that parameter one day to change in making your prognostication, isn't clinically enough. Reversing an anorectic (insert any animal) may take a few days to re-establish normal metabolism. And bring down cortisol levels. Remaining in the hospital likely is causing the level to remain elevated - so glucose alone would not be my main prognosticator.

Author Response 3:

We sincerely thank the reviewer for the in-depth and clinically insightful comments regarding anorexia, hepatic lipidosis, metabolic shifts, and the interpretation of blood glucose as a prognostic indicator in hedgehogs. We fully agree that blood glucose levels alone—especially measured over a short timeframe—are insufficient to determine prognosis in anorectic animals, and that nutritional support plays a critical role in improving outcomes by reversing hepatic lipidosis and restoring metabolic homeostasis.

As the reviewer correctly pointed out, relying solely on changes in glucose levels over one day is not clinically adequate. However, in our study, the term “next test” may have been misleading. While it could be interpreted by readers as referring to the following day, in reality, the follow-up glucose tests were conducted with a mean interval of 14.6 days (range, 3–30 days) after the initial examination. This has now been explicitly stated in the revised manuscript to avoid confusion. (Lines 407-416)

Furthermore, all hedgehogs in this study were treated as outpatients and were not hospitalized between visits. This decision was made with consideration for the species’ well-documented sensitivity to environmental stress. Therefore, the potential confounding effect of stress-induced hyperglycemia due to hospitalization was not present in these cases.

We fully acknowledge that normalization of blood glucose does not universally improve survival in all disease conditions. In the revised discussion, we have added a clear statement that intuitive clinical signs—such as improvements in appetite, overall condition, and resolution of presenting symptoms—should also be taken into account when evaluating prognosis. (Lines 416-431)

Given the scarcity of published clinical data on severe hyperglycemia in hedgehogs, this retrospective study aimed to provide preliminary insight into prognostic patterns observed in real-world clinical settings. While we recognize its limitations, we hope the findings may serve as a reference point for future studies and assist in developing more standardized clinical approaches.

Reviewer Comment 4:

I would not say this for pet hedgehogs. With my own, they nibble all night long, and many have a meal right before the daytime sleeping. They go into the day with a full stomach. Wild hedgehogs also scavenge during the night, and my experience with them is they have a relatively full stomach before sleep. So they aren't really fasted unless all your blood draws are confirmed on an empty stomach and done late afternoon/early evening.

Author Response 4:

We sincerely appreciate the reviewer’s thoughtful and experience-based comment regarding the feeding patterns of pet hedgehogs. We fully agree that it is not appropriate to assume a fasted state based solely on the nocturnal nature of hedgehogs or on the timing of blood collection during daytime hours. We acknowledge that many pet hedgehogs tend to nibble throughout the night and may even consume food shortly before entering their daytime sleep cycle, as also observed in wild hedgehogs.

Although exotic animal appointments at our clinic are generally scheduled in the afternoon rather than the morning, we understand that such details are not evident to readers and do not eliminate uncertainty regarding the feeding status of the animals at the time of blood collection. In response to the reviewer’s valuable input, we have revised the manuscript to more cautiously and clearly state the uncertainty regarding fasting status.

Additionally, we emphasize in the revised discussion that the 28 hedgehogs with severe hyperglycemia—the main focus of this study—included 24 (86%) individuals with markedly reduced appetite at the time of presentation. This clinical observation suggests that the blood glucose levels in these cases were more likely measured in a fasted or near-fasted physiological state rather than postprandially.

Nevertheless, we acknowledge as a key limitation of this retrospective study that the exact timing of feeding or fasting could not be confirmed from the medical records. This has been clearly stated in the revised manuscript. We agree that future prospective studies, with controlled feeding conditions and defined fasting periods, are essential for establishing more accurate and standardized reference values for blood glucose in hedgehogs. (Lines 432-441)

Reviewer Comment 5:

back to diets fed to pet hedgehogs. We have had DM hedgies, fed inappropriate diets. Most corrected when the diet was corrected, although if obese, a few may continue with likely type 2 diabetes. I have seen a juvenile onset DM (type 1) in color morphs that are highly inbred, however these too were not on an appropriate diet. Please discuss how pet hedgehog diets and obesity (some of the weight ranges you give place them into the obese category, particularly if female) is going to alter even serial blood glucose levels.

Author Response 5:

We sincerely thank the reviewer for sharing valuable clinical insights and for pointing out the importance of diet and obesity in the context of hyperglycemia and diabetes mellitus in pet hedgehogs. The case of juvenile-onset diabetes (suspected type 1) in highly inbred color morphs and the observation of dietary correction resolving hyperglycemia in many individuals were especially thought-provoking, and we deeply respect your clinical experience.

In response to your comment, we reviewed the medical records of the hedgehogs included in our study to assess the type of diets being fed. Most of the animals were reported to have been fed commercially available cat food or hedgehog-specific diets as their main staple. While inappropriate feeding was not identified in any case, a few individuals with severe anorexia were temporarily offered assisted diets such as liquid feline nutrition or insect-based food. However, because this dietary variation was limited to a small subset of individuals and occurred in the context of critical illness, we believe it is unlikely to have significantly influenced the overall results. The dietary information of hedgehogs with severe hyperglycemia was largely similar, and no association with any distinctive dietary management was identified. (Lines 337-339)

We also appreciate your important observation regarding the role of obesity in altering blood glucose levels. In our study, female hedgehogs with severe hyperglycemia had higher body weights than the reported normal range for the species, suggesting a tendency toward obesity. Additionally, based on your insightful suggestion, we reanalyzed the data by comparing survivors and non-survivors within the group of hedgehogs with severe hyperglycemia. This analysis revealed that non-survivors had significantly lower mean body weight and were older than survivors (p = 0.032 and p = 0.008, respectively), while there was no significant difference in initial blood glucose levels between the two groups (p = 0.523). This finding suggests that obesity may be a contributing factor in the development of hyperglycemia. However, in cases where hyperglycemia arises secondary to severe illness, obesity is less likely to act as a negative prognostic factor for survival. Indeed, lower body weight and older age are well-established risk factors for poor prognosis in many species. (Lines 179-181), (Table 8), (Lines 245-252), (Lines 337-358)

While the present study aimed to provide foundational clinical data on hyperglycemia in pet hedgehogs, we fully agree with the reviewer’s emphasis on the importance of further investigation into the role of diet, obesity, and diabetes classification. These variables remain key considerations in the clinical management of hedgehogs and warrant deeper exploration in future studies.

Reviewer Comment 6:

I would say "as a limited predictor"

Author Response 6:

We sincerely thank the reviewer for the valuable comment. We fully agree with the reviewer’s suggestion that assessing the severity of anorexia on the day of severe hyperglycemia diagnosis and determining recovery status at follow-up should not be regarded as strong or definitive prognostic indicators. Rather, they should be interpreted as limited and supportive factors that require cautious clinical application.

Accordingly, we have revised the manuscript to describe these indicators as a "potential predictor" and a "potential and supportive prognostic indicator" respectively. These expressions were chosen to clarify that, while the indicators may have clinical relevance, they are not robust predictors on their own and should be interpreted in the context of other clinical signs such as appetite improvement, general condition, and resolution of presenting symptoms. These findings provide preliminary insights that may aid early prognosis and treatment planning in hyperglycemic hedgehogs, though their use should remain cautious and contextual.

(Lines 21-23), (Lines 34-37), (Lines 385-406),(Lines 416-431),(Lines 455-461)

We believe that the reviewer’s insights have helped us improve the clarity and clinical applicability of the manuscript, and we are grateful for the guidance.

4. Response to Comments on the Quality of English Language

Point 1: The English is fine and does not require any improvement.

Response 1: We sincerely appreciate your positive feedback on the quality of English used in the manuscript.

5. Additional clarifications

We would like to express our sincere gratitude to the reviewer for the thoughtful, detailed, and insightful comments provided throughout the review process. Your feedback has greatly contributed to improving the clarity, rigor, and scientific value of our manuscript.

While we have made every effort to address all comments thoroughly, we would especially appreciate your further assessment regarding one remaining concern—the inclusion of the healthy hedgehog group.

To compare the previously reported reference blood glucose value of 89 ± 30 mg/dL with the mean glucose level observed in our study’s healthy hedgehog group, we selected healthy individuals using a new set of criteria that were independent of those applied to the other groups. Details on this group are provided in the Materials and Methods section (2.2. Healthy Hedgehog Criteria), the Results section (3.2. Healthy Hedgehog Group), and the relevant part of the Discussion.

However, we acknowledge that our original descriptions may have unintentionally suggested that this group was established for direct statistical comparison with the severe hyperglycemia group. To clarify this potential source of confusion, we have added explicit statements in all relevant sections stating that the healthy group was constructed for descriptive purposes only and was not intended as a statistical control. If this clarification is still considered insufficient or potentially misleading, we would be grateful for further guidance. We are fully prepared to remove this subset from the manuscript if necessary, and we believe that the integrity and core findings of the study would remain intact even without it.

In addition, we would like to provide the following clarifications:

  • Terminology Adjustments: In the revised manuscript, we have clarified the use of the terms "frequency" and "prevalence" in accordance with the level of analysis (sample-level vs. individual-level). This distinction was made to ensure terminological accuracy and to better reflect the structure of our dataset.
  • Structure of the Results Section: To avoid misinterpretation, we have added a statement at the beginning of the Results section emphasizing that the findings are based on three independent datasets with distinct inclusion criteria. This clarification aims to prevent readers from assuming statistical comparability between the groups.
  • Line Number Referencing: All major revisions and additions have been highlighted using track changes in the resubmitted Word manuscript. In the resubmitted PDF file, line numbers have been referenced throughout the response document to assist reviewers and editors in locating the changes efficiently.

Once again, we are deeply grateful for your guidance. Your comments helped us identify and revise elements that could have caused confusion for readers, and we truly appreciate your contribution to improving the transparency and scientific quality of our work.

Reviewer 3 Report

Comments and Suggestions for Authors

This manuscript tackles a relevant yet underexplored topic in exotic animal medicine—the occurrence and clinical profile of hyperglycemia in pet African pygmy hedgehogs. The authors should be commended for bringing attention to a condition that is seldom documented in the literature, and for attempting to identify prognostic indicators that may guide clinical management.

That said, the study's findings would benefit significantly from a more rigorous and structured statistical approach. At present, the results are primarily descriptive, and while they offer some insight into the prevalence and presentation of hyperglycemia, they fall short of establishing meaningful associations between clinical variables and glucose levels.

Key factors such as age, body weight, underlying disease, and outcome are mentioned, but no statistical tests are applied to evaluate how these variables relate to hyperglycemia severity. Without such analysis, it is difficult to draw reliable conclusions or to understand which patient profiles are at greater risk.

To strengthen the study, I would recommend incorporating, for example, group comparisons (e.g., ANOVA or non-parametric equivalents) where appropriate.

Moreover, potential confounders should be addressed explicitly, particularly given the complexity of metabolic disturbances in small exotic mammals.

In summary, this article is a promising first step in characterizing hyperglycemia in African pygmy hedgehogs, but the clinical value of the findings remains limited without a more thorough statistical evaluation. A deeper analytical layer would not only enhance the robustness of the conclusions but also make the study more relevant for both clinical application and future research

Author Response

For research article

Response to Reviewer X Comments

1. Summary

We would like to sincerely thank the reviewer for the time and effort dedicated to evaluating our manuscript. We greatly appreciate the thoughtful, constructive, and detailed comments, which have significantly contributed to improving the clarity, accuracy, and scientific quality of our work. In response, we have revised the manuscript thoroughly and addressed each point raised by the reviewer. All corresponding changes have been clearly indicated using track changes in the resubmitted version of the manuscript. Below, we provide a point-by-point response to each comment.

2. Questionsfor General Evaluation

Reviewer’s Evaluation

Response and Revisions

Does the introduction provide sufficient background and include all relevant references?

Yes/Can be improved/Must be improved/Not applicable

Are all the cited references relevant to the research?

Yes/Can be improved/Must be improved/Not applicable

Is the research design appropriate?

Yes/Can be improved/Must be improved/Not applicable

Are the methods adequately described?

Yes/Can be improved/Must be improved/Not applicable

Are the results clearly presented?

Yes/Can be improved/Must be improved/Not applicable

Are the conclusions supported by the results?

Yes/Can be improved/Must be improved/Not applicable

3. Point-by-point response to Comments and Suggestions for Authors

Comments 1:

This manuscript tackles a relevant yet underexplored topic in exotic animal medicine—the occurrence and clinical profile of hyperglycemia in pet African pygmy hedgehogs. The authors should be commended for bringing attention to a condition that is seldom documented in the literature, and for attempting to identify prognostic indicators that may guide clinical management.

That said, the study's findings would benefit significantly from a more rigorous and structured statistical approach. At present, the results are primarily descriptive, and while they offer some insight into the prevalence and presentation of hyperglycemia, they fall short of establishing meaningful associations between clinical variables and glucose levels.

Key factors such as age, body weight, underlying disease, and outcome are mentioned, but no statistical tests are applied to evaluate how these variables relate to hyperglycemia severity. Without such analysis, it is difficult to draw reliable conclusions or to understand which patient profiles are at greater risk.

To strengthen the study, I would recommend incorporating, for example, group comparisons (e.g., ANOVA or non-parametric equivalents) where appropriate.

Moreover, potential confounders should be addressed explicitly, particularly given the complexity of metabolic disturbances in small exotic mammals.

In summary, this article is a promising first step in characterizing hyperglycemia in African pygmy hedgehogs, but the clinical value of the findings remains limited without a more thorough statistical evaluation. A deeper analytical layer would not only enhance the robustness of the conclusions but also make the study more relevant for both clinical application and future research

Response 1:

Thank you very much for your thoughtful and constructive comment. We fully agree that structured statistical analysis plays a key role in identifying clinically meaningful associations and enhancing the interpretability of observational studies.

We would like to take this opportunity to explain the rationale behind our study design and analytical approach.

This study was based on three independent datasets, each constructed with different objectives and inclusion criteria:

1. Blood Glucose Frequency Dataset (579 blood samples):

This dataset includes all blood glucose values collected from 252 hedgehogs, including repeated measurements from the same individuals. As the analysis was conducted at the sample level, not the individual level, we have revised the manuscript to replace the term prevalence with frequency. The purpose of this dataset was to descriptively evaluate the overall distribution of glucose concentrations across the entire study population.

2. Healthy Hedgehog Group (n = 80):
This group was constructed using the earliest available glucose value from hedgehogs that were clinically healthy at the time of routine wellness examinations. Data on age, body weight, blood glucose level, and sex were collected to serve as a reference for descriptive comparison with previously reported normal glucose ranges (e.g., 89 ± 30 mg/dL). As this group was not randomly selected or matched, it was not used as a statistical control group for inferential analysis.

3. Severe Hyperglycemia Group (n = 28):
This group included hedgehogs with an initial blood glucose concentration of ≥180 mg/dL at the time of presentation. We collected information on age, sex, body weight, glucose level, major clinical signs, and survival outcome to descriptively explore potential prognostic indicators associated with severe hyperglycemia.

Because the three groups differed in terms of purpose, selection strategy, and clinical context, we determined that direct statistical comparisons were not appropriate. Essential assumptions such as randomization, independence, and comparability could not be satisfied. In particular, constructing a valid control group for the severe hyperglycemia cohort would have required matching for variables such as age, time of presentation, and comorbidities—an approach that was not feasible due to the retrospective nature of the dataset and variability in clinical presentation.

In response to the reviewer’s suggestion, we also explored the distribution of sex between the healthy hedgehog group and the severe hyperglycemia group using the chi-square test. The result indicated no statistically significant difference (p = 0.844). However, as the healthy group was not randomly selected but rather constructed under specific inclusion criteria (i.e., based on the earliest available glucose measurement in clinically healthy individuals), we consider it methodologically inappropriate to treat this group as a valid comparator for inferential analysis. For the same reason, variables such as age and body weight were not compared between the two groups, in order to avoid potential bias introduced by non-random selection.

To address these limitations, future studies should adopt prospective designs or matched case-control frameworks. We agree that such approaches would better elucidate the clinical associations between blood glucose levels and prognostic variables in hedgehogs.

We also appreciate your important observation regarding the role of obesity in altering blood glucose levels. In our study, female hedgehogs with severe hyperglycemia had higher body weights than the reported normal range for the species, suggesting a tendency toward obesity. Additionally, based on your insightful suggestion, we reanalyzed the data by comparing survivors and non-survivors within the group of hedgehogs with severe hyperglycemia. This analysis revealed that non-survivors had significantly lower mean body weight and were older than survivors (p = 0.032 and p = 0.008, respectively), while there was no significant difference in initial blood glucose levels between the two groups (p = 0.523). This finding suggests that obesity may be a contributing factor in the development of hyperglycemia. However, in cases where hyperglycemia arises secondary to severe illness, obesity is less likely to act as a negative prognostic factor for survival. Indeed, lower body weight and older age are well-established risk factors for poor prognosis in many species.

We also recognize that, despite being analyzed independently, the presentation of the healthy and severe hyperglycemia groups in the original manuscript may unintentionally imply comparability. To address this and improve clarity, we have added the following statement at the beginning of the Results section:

“To improve clarity, we note that the results are structured around three separate datasets, each with unique inclusion criteria and analytical purposes. As such, they represent independent study groups and are not intended for direct intergroup comparison.”

We hope that this addition will help readers clearly understand the structure and scope of each dataset and interpret the results accordingly within their respective clinical contexts.

Thank you for your insightful comment regarding the descriptive nature of our results. In response, we have revised the Discussion section extensively (Lines 259–451) to enhance clarity and contextual interpretation. We have added more relevant references and expanded upon them to support our interpretations more robustly.

As you correctly pointed out, our study lacked a statistically comparable control group for the severe hyperglycemia cohort. We fully acknowledge this limitation, and we now explicitly emphasize that the findings should be interpreted with caution. To address this, we elaborated on the clinical relevance of the results while clearly stating the need for cautious interpretation in the absence of matched controls.

In the original manuscript, the 579 blood samples were derived from 252 individual hedgehogs, including multiple measurements from some individuals. Therefore, we revised the terminology: the term "prevalence" was replaced with "frequency" when describing glucose distribution at the sample level. However, for the reported 11.1% prevalence of severe hyperglycemia (based on one sample per individual across 252 hedgehogs), the term "prevalence" was retained as appropriate.

Additionally, in response to your comment that “potential confounders should be addressed explicitly, particularly given the complexity of metabolic disturbances in small exotic mammals,” we have expanded the Introduction (Lines 44–56) to include factors known to influence blood glucose levels in small mammals. This includes fasting duration, type of diet, anesthesia-related stress, reproductive status, and obesity.

We also strengthened the Discussion throughout (Lines 296–441) to reflect these confounding factors in more detail. Specifically, we addressed the potential effects of isoflurane anesthesia, transport-related stress, feeding status in wild versus pet hedgehogs, obesity, and stress-induced hyperglycemia associated with physical abnormalities, pain, or underlying disease. We further discussed observed patterns regarding body weight and hyperglycemia, age-related trends, dietary composition, and sex distribution.

Your insightful comment greatly improved the clarity of the manuscript and guided us to perform additional subgroup analyses, which revealed new clinical insights related to body weight and age. We sincerely appreciate your thoughtful feedback, which strengthened both the depth and quality of this study.

4. Response to Comments on the Quality of English Language

Point 1: The English is fine and does not require any improvement.

Response 1: We sincerely appreciate your positive feedback on the quality of English used in the manuscript.

5. Additional clarifications

We would like to express our sincere gratitude to the reviewer for the thoughtful, detailed, and insightful comments provided throughout the review process. Your feedback has greatly contributed to improving the clarity, rigor, and scientific value of our manuscript.

While we have made every effort to address all comments thoroughly, we would especially appreciate your further assessment regarding one remaining concern—the inclusion of the healthy hedgehog group.

To compare the previously reported reference blood glucose value of 89 ± 30 mg/dL with the mean glucose level observed in our study’s healthy hedgehog group, we selected healthy individuals using a new set of criteria that were independent of those applied to the other groups. Details on this group are provided in the Materials and Methods section (2.2. Healthy Hedgehog Criteria), the Results section (3.2. Healthy Hedgehog Group), and the relevant part of the Discussion.

However, we acknowledge that our original descriptions may have unintentionally suggested that this group was established for direct statistical comparison with the severe hyperglycemia group. To clarify this potential source of confusion, we have added explicit statements in all relevant sections stating that the healthy group was constructed for descriptive purposes only and was not intended as a statistical control. If this clarification is still considered insufficient or potentially misleading, we would be grateful for further guidance. We are fully prepared to remove this subset from the manuscript if necessary, and we believe that the integrity and core findings of the study would remain intact even without it.

In addition, we would like to provide the following clarifications:

  • Terminology Adjustments: In the revised manuscript, we have clarified the use of the terms "frequency" and "prevalence" in accordance with the level of analysis (sample-level vs. individual-level). This distinction was made to ensure terminological accuracy and to better reflect the structure of our dataset.
  • Structure of the Results Section: To avoid misinterpretation, we have added a statement at the beginning of the Results section emphasizing that the findings are based on three independent datasets with distinct inclusion criteria. This clarification aims to prevent readers from assuming statistical comparability between the groups.
  • Line Number Referencing: All major revisions and additions have been highlighted using track changes in the resubmitted Word manuscript. In the resubmitted PDF file, line numbers have been referenced throughout the response document to assist reviewers and editors in locating the changes efficiently.

Once again, we are deeply grateful for your guidance. Your comments helped us identify and revise elements that could have caused confusion for readers, and we truly appreciate your contribution to improving the transparency and scientific quality of our work.

Round 2

Reviewer 1 Report

Comments and Suggestions for Authors

Thank you for your thorough responses to the comments. I have a couple of minor concerns.

Line 74: I think indicative is the wrong word here since hyperglycemia is defined as elevated blood glucose.

Line 484: I think you meant that the appetite is low.

Regarding the discussion, there is still a lot of repetition of the actual results. Ex: Lines 514-516 repeat the data already presented. The rest of the paragraph is a good discussion of the usefulness and limitations of the findings. I'll defer to the editor, but a summary sentence of your findings, rather than repeating the numbers would make the discussion read a little better.

Author Response

Reviewer Comment (Line 74):

I think "indicative" is the wrong word here since hyperglycemia is defined as elevated blood glucose.

Author Response:

Thank you for your valuable feedback. We agree with your observation that the phrase "indicative of elevated blood glucose levels" is redundant, as hyperglycemia is by definition a state of elevated blood glucose. In response, we revised the sentence to improve both accuracy and clarity.

Additionally, to avoid repetitive use of the “not only... but also...” construction in two consecutive sentences, we also revised the preceding sentence for better flow and stylistic variety.

The revised sentences now read as follows:  (Lines 54–56)

“The term stress hyperglycemia is used to describe a condition where hyperglycemia is induced by illness or fear in patients without diabetes. This phenomenon is observed in both humans and animals, including dogs, cats, and horses [17–20]. In veterinary medicine, stress hyperglycemia is considered not only a physiological response but also a potential prognostic indicator of disease outcomes.

We believe these revisions address the redundancy you pointed out while improving the overall readability and logical progression of the paragraph. Thank you for helping us strengthen the manuscript.

Reviewer Comment (Line 484):

I think you meant that the appetite is low.

Author Response:

(As the content mentioned in Reviewer’s Line 484 corresponds to Lines 386–388 in our PDF manuscript, we would like to provide our response based on that section.)

Thank you very much for your careful reading and insightful suggestion.

We were unable to locate Line 484 in our submitted manuscript due to differences in line numbering between versions. Therefore, we have provided our response using the line numbers from the PDF version of the revised manuscript.

Upon reviewing the content, we identified the relevant sentence and revised it accordingly for clarity and accuracy. Additionally, we found two other nearby expressions that could benefit from improved wording and made the following changes:

  1. Lines 386–388:
    Original: “The results of this study showed that the mortality rate was higher when the anorexia was low on the day of examination.”
    Revised: “The results of this study showed that the mortality rate was higher when the severity of anorexia was greater on the day of examination.”
  2. Lines 388–389:
    Original: “These results suggest that the degree of anorexia on the day when severe hyperglycemia was first identified can be used as a useful early warning signal to predict the prognosis of hedgehogs.”
    Revised: “These results suggest that the severity of anorexia on the day when severe hyperglycemia was first identified can be used as a useful early warning signal to predict the prognosis of hedgehogs.”
  3. Lines 404–405:
    Original: “The severity of appetite is practical for clinicians in that it is an easily observable factor.”
    Revised:Assessing the severity of anorexia is practical for clinicians, as it is an easily observable factor.

We sincerely appreciate your detailed feedback. Without your suggestion, these expressions may have remained ambiguous to readers. Your comment significantly improved the clarity and precision of our manuscript.

Reviewer Comment:
Regarding the discussion, there is still a lot of repetition of the actual results. Ex: Lines 514–516 repeat the data already presented. The rest of the paragraph is a good discussion of the usefulness and limitations of the findings. I'll defer to the editor, but a summary sentence of your findings, rather than repeating the numbers would make the discussion read a little better.

Author Response:
Thank you very much for your insightful comment. We agree that the numerical values mentioned in Lines 514–516 were previously presented in the Results section. However, we respectfully decided to retain this sentence in the Discussion to emphasize the magnitude of the mortality difference between the recovered and non-recovered groups. We believe that explicitly stating this contrast (8.3% vs. 87.5%), together with its statistical significance (p < 0.001), helps to reinforce the clinical importance of follow-up recovery status as a potential prognostic indicator.
We would also like to sincerely thank you for your detailed and thoughtful feedback throughout the review process. Your careful attention to the manuscript greatly helped us refine our reasoning and improve the clarity of our work. We truly appreciate your contribution to enhancing the quality of this paper.

Reviewer 2 Report

Comments and Suggestions for Authors

My concerns have been addressed. 

Author Response

Reviewer 2 Comment:
My concerns have been addressed.

Author Response:
Thank you very much for your kind review and confirmation. We sincerely appreciate your time and thoughtful feedback throughout the review process. Your comments were extremely helpful in improving the clarity and overall quality of our manuscript.

Reviewer 3 Report

Comments and Suggestions for Authors

I would like to thank the authors for their efforts and for carefully addressing all the comments and suggestions. It is evident that they have made a substantial effort to improve the quality of the manuscript, and the revisions significantly enhance its scientific and structural value. I consider the current version suitable for publication.

Author Response

Reviewer 3 Comment:
I would like to thank the authors for their efforts and for carefully addressing all the comments and suggestions. It is evident that they have made a substantial effort to improve the quality of the manuscript, and the revisions significantly enhance its scientific and structural value. I consider the current version suitable for publication.

Author Response:
We sincerely thank you for your generous and encouraging comments. We truly appreciate your thoughtful review, which motivated us to carefully refine and improve the scientific rigor and structure of the manuscript. Your kind words and support mean a great deal to us.